# Global genomic analyses of wheat powdery mildew reveal association of pathogen spread with historical human migration and trade

Alexandros G. Sotiropoulos [1✉], Epifanía Arango-Isaza [2], Tomohiro Ban[3], Chiara Barbieri[2,4], Salim Bourras [1,5], Christina Cowger[6], Paweł C. Czembor [7], Roi Ben-David [8], Amos Dinoor[9], Simon R. Ellwood [10], Johannes Graf [1], Koichi Hatta[11], Marcelo Helguera [12], Javier Sánchez-Martín [1], Bruce A. McDonald[13], Alexey I. Morgounov [14], Marion C. Müller [1], Vladimir Shamanin[15], Kentaro K. Shimizu [2,3], Taiki Yoshihira[16], Helen Zbinden[1], Beat Keller [1] & Thomas Wicker [1✉]

The fungus *Blumeria graminis* f. sp. *tritici* causes wheat powdery mildew disease. Here, we study its spread and evolution by analyzing a global sample of 172 mildew genomes. Our analyses show that *B.g. tritici* emerged in the Fertile Crescent during wheat domestication. After it spread throughout Eurasia, colonization brought it to America, where it hybridized with unknown grass mildew species. Recent trade brought USA strains to Japan, and European strains to China. In both places, they hybridized with local ancestral strains. Thus, although mildew spreads by wind regionally, our results indicate that humans drove its global spread throughout history and that mildew rapidly evolved through hybridization.

[1] Department of Plant and Microbial Biology, University of Zurich, Zurich, Switzerland. [2] Department of Evolutionary Biology and Environmental Studies, University of Zurich, Zurich, Switzerland. [3] Kihara Institute for Biological Research, Yokohama City University, Yokohama, Kanagawa, Japan. [4] Department of Linguistic and Cultural Evolution, Max Planck Institute for Evolutionary Anthropology, Leipzig 04103, Germany. [5] Department of Forest Mycology and Plant Pathology, Swedish University of Agricultural Sciences, Uppsala, Sweden. [6] USDA-ARS Department of Plant Pathology, North Carolina State University, Raleigh, NC, USA. [7] Plant Breeding and Acclimatization Institute - National Research Institute, Radzików, 05-870 Błonie, Poland. [8] Department of Vegetables and Field crops, Institute of Plant Sciences, ARO-Volcani Center, Rishon LeZion 7528809, Israel. [9] Department of Plant Pathology and Microbiology, The Robert H. Smith Faculty of Agriculture, Food & Environment, The Hebrew University of Jerusalem, Rehovot, Israel. [10] Centre for Crop and Disease Management, School of Molecular and Life Sciences, Curtin University, Bentley, WA 6102, Australia. [11] Hokkaido Agricultural Research Center Field Crop Research and Development, National Agricultural Research Organization, Sapporo, Hokkaido, Japan. [12] Centro de Investigaciones Agropecuarias (CIAP), INTA, Córdoba, Argentina. [13] Plant Pathology, Institute of Integrative Biology, ETH Zurich, Zurich, Switzerland. [14] Food and Agriculture Organization of the United Nations, Riyadh, Saudi Arabia. [15] Omsk State Agrarian University, Omsk, Russia. [16] Department of Sustainable Agriculture, Rakuno Gakuen University, Ebetsu, Hokkaido, Japan. ✉email: alexandrosgsotiropoulos@gmail.com; wicker@botinst.uzh.ch

*B*lumeria graminis is a fungal plant pathogen that causes powdery mildew, the eighth most economically damaging disease in wheat[1]. It can lead to estimated annual yield losses of 7.6–19.9%[2,3]. The species *B. graminis* contains multiple *formae speciales*, defined by their host specificity, e.g., *Blumeria graminis forma specialis tritici* (*B.g. tritici*) is specialized to infect hexaploid (bread) wheat, while *B.g. dicocci* grows mostly on tetraploid wheat. *B. graminis* has a sexual life cycle leading to chasmothecia formation to survive harsh conditions and an asexual (disease) cycle. The 166 Mb genome of wheat powdery mildew was recently sequenced and assembled into full chromosomes[4].

Wheat was domesticated in the Fertile Crescent ~10,500 years ago[5,6]. From there, it spread throughout Eurasia, playing a crucial role in the development of agriculture and civilization. In the Americas, wheat was introduced by Europeans during the 16th to 17th century[7]. Intensifying trade established trade routes and exchange of wheat varieties worldwide. During the last century, countries like the USA or Russia became major exporters of wheat[8], while others such as Japan, where wheat was introduced ~2000 years ago, received large imports from the USA[9].

Human activities have long been associated with the spread of pathogens, as demonstrated by the COVID-19 pandemic[10]. The complex breeding and trade history of wheat raises questions on how wheat diseases spread, adapted, and co-evolved with their hosts. Barley powdery mildew was shown to spread by wind up to ~100 km per year[11]. However, it is not known whether wind can transport mildew spores between continents, or whether humans are required as vectors, as was shown e.g. for chestnut blight, coffee rust, Dutch elm, wheat blast, wheat yellow rust, and *Fusarium oxysporum* disease fungi[12–19].

Crop breeding can also drive the emergence of new pathogen strains[20]. For example, the introduction of the hybrid crop triticale was rapidly mirrored by the hybridization of mildews of its two parent species[21]. That study also proposed that wheat mildew itself is an ancient hybrid of mildews that grew on tetraploid or diploid wheat progenitors. Hybridization can even occur between barley and wheat mildew *formae speciales*, which diverged millions of years ago[22], demonstrating that hybridization is possible between very distantly related lineages. For simplicity, we hereafter refer to recombination between mildew lineages as "hybridizations", regardless of whether they occur between *formae speciales* or between lineages of the same *forma specialis*, which diverged hundreds or thousands of years ago.

Previous population genomics studies on global or regional samples of plant pathogenic fungi (e.g., *Calonectria spp.*, *Cryphonectria parasitica*, *Hemileia vastatrix*, *Hymenoscyphus fraxineus*, *Ophiostoma spp.*, *Magnaporthe oryzae*, *Parastagonospora nodorum*, *Puccinia striiformis*, *Puccinia triticina*, *Rhynchosporium commune*, and *Zymoseptoria tritici*[19,23–37]) explored different aspects of pathogen evolution. Some used reduced representation data such as SSRs and RADseq[24,29,31], while others used whole-genome sequencing technologies[19,26,27,33,35–38]. Furthermore, multiple studies suggested the hybridization of different pathogen lineages as drivers of pathogen evolution[12,19,21,30,31,36]. Analyses of powdery mildew pathogens were so far limited to isolates representing relatively small geographical regions[39–41], and there were, to our knowledge, only a few studies on worldwide samples of obligate plant biotrophs using whole-genome sequencing (e.g., *P. triticina*[27]).

In this work, we analyse the genomes of a worldwide sample of 172 wheat powdery mildew isolates. Our data indicate that wheat powdery mildew originated in the Fertile Crescent, from which it spread through Europe and eventually other continents. Importantly, we found evidence that wheat powdery mildews hybridized multiple times with other mildew strains, leading to distinct lineages in America, Japan, and China. We conclude that human migration and trade during pre-historic and historic times were major drivers of the spread of this important wheat pathogen.

## Results and discussion

We compiled a collection of 172 *B. tritici/dicocci* isolates from 13 countries and five continents in order to cover the species range (Fig. 1a, Supplementary Data 1, Methods). A total of 76 isolates were sequenced for this study, 69 from *B.g. tritici* and 7 from *B.g. dicocci*, and newly generated genomes were merged with published data[21,42,43], including 12 isolates of *B.g. dicocci* (i.e., tetraploid wheat mildew from Israel)[21]. In all geographical populations with at least eight individuals, both mating types were found not statistically deviating from the expected 50/50 ratio, suggesting sexual recombination within populations (Supplementary Tables 1 and 2).

Genomes were sequenced from 20 to 167-fold coverage, with an average of 45-fold (Supplementary Fig. 1). We used GATK for SNP calling, resulting in a primary dataset of 2,475,870 quality filtered SNPs, of which 2,146,243 had a genotyping rate (i.e., the proportion of isolates for which SNP data were available) of at least 90%. Most SNPs were present in low frequencies, with ~42.4% of the polymorphic positions being singletons. Approximately 24.9% had a minor allele frequency >0.05, a minimum depth (i.e., number of reads that cover an SNP) of 3, and a maximum of 1000. SNP quality and alternate allele frequency correlated strongly, indicating sufficient detection of alternate alleles (Supplementary Fig. 2, Methods).

Deleterious/high impact mutations showed the strongest bias toward rare alleles, indicating that selection prevented these SNPs from spreading through populations (Supplementary Fig. 3). Furthermore, we detected 112,465 insertions or deletions (InDels), ranging from 1 to 296 bp, with the majority being at low frequency (Supplementary Fig. 3). Most InDels (43.5%) were 1 bp in size, and those present in coding regions were strongly biased toward lengths that are multiples of three, reflecting purifying selection to maintain reading frames (Supplementary Fig. 4).

**Wheat powdery mildew comprises multiple distinct populations.** To infer population subdivision, we calculated simple mismatch distances between individual isolates and performed a principal coordinate analysis (PCoA). To add robustness to the analysis, we also performed PCoA using Bray–Curtis and Nei's genetic distances, which yielded very similar results (Supplementary Fig. 5). The PCoA of the 172 isolates showed clear clustering of most geographical groups. The first principal coordinate (PCo1) explained 16.5% of the variance and separated groups mostly by geographical origins (Fig. 1b and Supplementary Fig. 5). The second coordinate (PCo2) separated *B.g. dicocci* from the *B.g. tritici* isolates. This was expected as *B.g. tritici* is a different *forma specialis*. When *B.g. dicocci* isolates were excluded, PCoA revealed the strong association between geographical origin and genetic proximity of the isolates (Fig. 1c). Populations from Europe, Middle Asia (RUS-KAZ), and the Fertile Crescent clustered closely. In contrast, populations from China, Argentina, USA, and Japan formed distinct groups. Additionally, mildew isolates from Australia and the USA are indistinguishable, suggesting a recent migration of the pathogen from the USA to Australia (see section "A model for the worldwide spread of wheat powdery mildew", Supplementary Note 1). However, this presumption must be taken with caution since only three Australian isolates were represented.

The presence of such distinct mildew populations raised the question of whether we could identify different ancestries in individual populations. Thus, ADMIXTURE analysis was performed

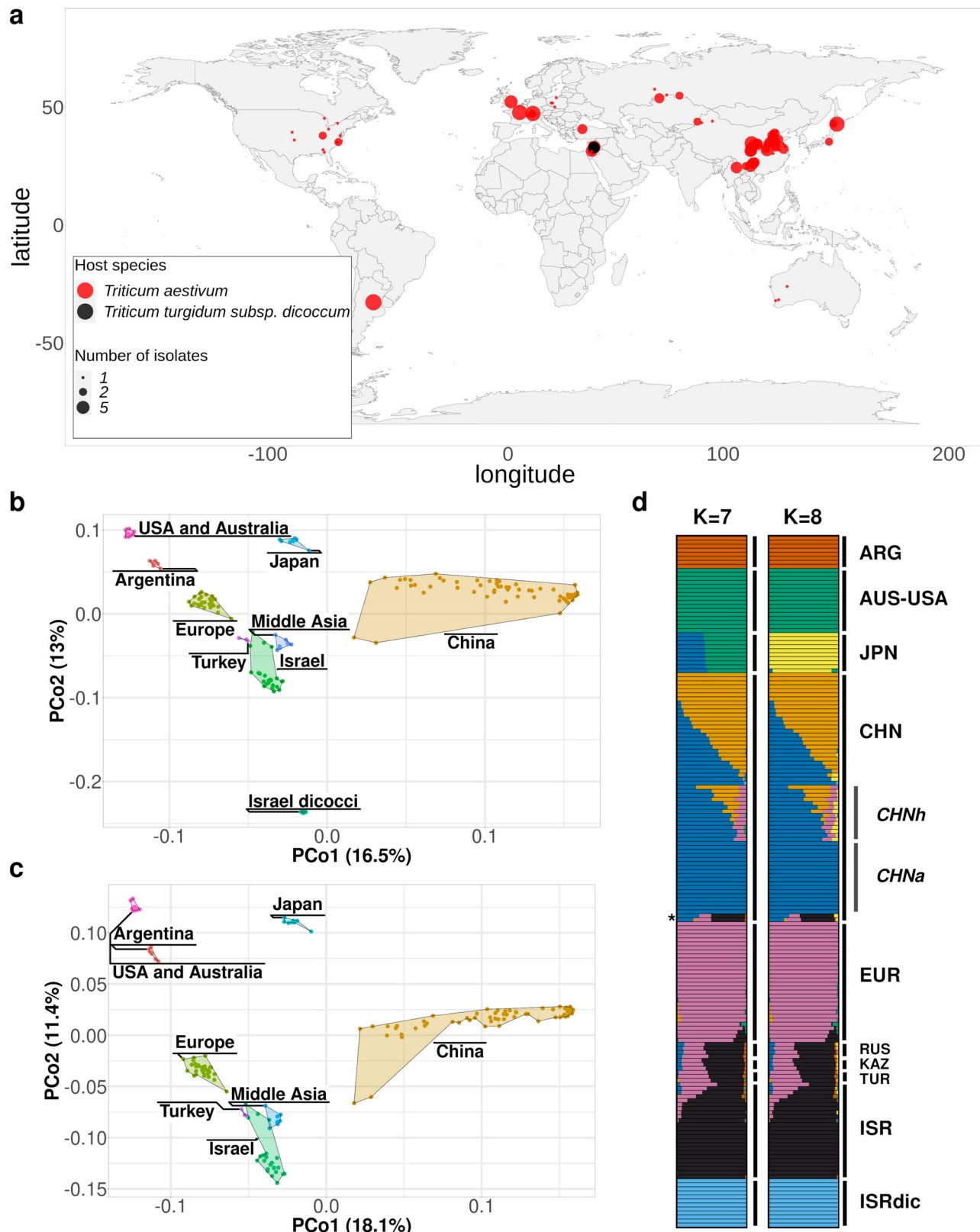

**Fig. 1 Basic information about the wheat powdery mildew isolates. a** World map with points where the isolates were collected from. The color corresponds to the host where the isolates were isolated and the size of the dot corresponds to the number of isolates collected at that place. **b** PCoA analysis plot using simple mismatch distances of LD-pruned SNPs of the 172 isolates with the principal components 1 and 2 including the *B.g. dicocci forma specialis*. **c** PCoA analysis plot using simple mismatch distances of LD-pruned SNPs of the 160 isolates with the principal components 1 and 2, excluding *B.g. dicocci*. **d** Admixture plot for ancestral populations with the two most probable numbers of K = 7 and K = 8 for the 172 isolates with LD-pruned SNPs with a minor allele frequency >0.1. The asterisk indicates two isolates from China with ancestry from the Fertile Crescent.

to explore degrees of population admixture with up to 15 ancestries. Cross-validation error indicated the optimal number of ancestral populations (K) was between seven and eight, with the most probable K = 7 (Fig. 1d, Supplementary Fig. 6). The *forma specialis B.g. dicocci* separates early from all others, at K = 3 because the large population of all Chinese mildew isolates [CHN], with its high geographic and genetic distances between individuals, was driving the first separation of ancestries at K = 2. All European, West, and Mid-Asian populations share part of an ancestry until K = 4. European and Mid-Asian ancestries separate from K = 5, while *B.g. tritici* isolates from Kazakhstan, Russia, Turkey and Israel show further admixture at higher K values. The American mildew isolates share a mostly unique ancestry for K = 4 and K = 5, above which the Argentinian isolates differentiate. The three Australian isolates share the same ancestry as the USA isolates for almost all values of K (Supplementary Fig. 6).

The ADMIXTURE plots were also suggestive of hybrid mildew strains. For example, an East Asian component predominates in most Chinese isolates (in blue in Fig. 1d and Supplementary Fig. 6), suggesting it comprised a largely independent gene pool. We referred to isolates with only the East Asian component CHNa ("a" for ancestral) and to those with more than 8% of the European ancestry as CHNh ("h" for putative hybrid). Additionally, two Chinese isolates shared a high proportion of ancestry with isolates from Israel (Fig. 1d and Supplementary Fig. 6). Moreover, all Japanese isolates shared ancestry with mildews from East Asia and the USA from K = 2 to K = 7. (Fig. 1d and Supplementary Fig. 6). This could be interpreted as the Japanese isolates being hybrids of mildews from USA and China (see section "Hybridizations in Japanese and Chinese mildews").

**Geographical and genetic distances of wheat powdery mildews are associated at large scales.** Because efficient wind dispersal at a regional level has been shown previously, we hypothesized that wind may also contribute to the spread of the pathogen across and between continents. Therefore, we studied the association of genetic and geographical distances in our worldwide dataset as well as within subsets, using the Mantel test and the distance-based Moran Eigenvector Maps analysis (dbMEM). Interestingly, the Mantel test showed strong correlations between genetic and geographical distances between isolates when broad datasets were used, such as between continents or across Eurasia, indicating that wind dispersal over large distances is not a major driver of pathogen spread (examples in Fig. 2a and Supplementary Fig. 7). However, we found hardly any significant correlation between geographical origin and genetic similarity within individual geographical mildew populations, supporting the previous findings[11]. Mildews from Japan and USA were exceptions, where sampling sites were sometimes separated by large geographical distances (examples in Fig. 2a and Supplementary Fig. 7). The more sensitive Moran Eigenvector Maps analysis (dbMEM) analysis supported all the population separations of the Mantel test (Supplementary Fig. 8). Additionally, the first eigenvector separated American, European and Israeli mildew isolates from the rest of Asian isolates, while the second showed Chinese mildews clustering with the American ones, separating from the rest of the Eurasian isolates (Supplementary Fig. 8).

Taken together, these data indicated there was a distinct grouping of geographical populations with some genetic exchange between mildew strains at a regional level, while large geographic distances (~>500 km) effectively discriminate mildew populations.

**The Fertile Crescent is the likely region of origin of wheat powdery mildew.** Since hexaploid wheat originated in the Fertile Crescent[5,6], we hypothesized that *B.g. tritici* also emerged there.

Thus, mildews from that region were expected to be most diverse. Indeed, the *B.g. tritici* mildew population from Israel had the highest sequence diversity π, the highest average differences within populations (ADW) using simple mismatch distances for LD-pruned and non-LD-pruned SNP datasets, as well as the highest Watterson's Theta, closely followed by European mildews (Fig. 2b, Supplementary Table 3, and Supplementary Fig. 9). Mildew populations from USA and Argentina had the lowest average nucleotide diversity and lowest numbers of segregating sites. Although all Argentinian isolates came from a single field and time point, their diversity was similar to that of isolates collected across the West and Midwest of the USA, suggesting that sampling bias was not a likely explanation for the low diversity. Instead, founder effects and/or genetic drift may have contributed to the low diversity.

To further investigate relationships between populations, distance-based networks were constructed using fourfold degenerate sites, alignments of all polymorphic nucleotide positions in CDS, and simple mismatch distances. Here, mildews from rye (*B.g. secalis*), barley (*B.g. hordei*) and wild grasses (*B.g. poae*, *B.g. avenae*, and *B.g. lolii*)[44] were included as outgroups. Overall topologies of all networks were consistent, with the different *formae speciales* proximal to *B.g. dicocci*, but distant from *B.g. tritici* populations (examples provided in Fig. 2c and Supplementary Figs. 10, 11). *B.g. tritici* isolates from Israel were in all networks positioned at the base of the *B.g. tritici* clade (Fig. 2c and Supplementary Figs. 10, 11), again suggesting the Fertile Crescent as the region of origin. In addition, mildews from America (the USA and ARG) and Japan formed separate branches of the network, with Japanese mildews having connections to mildews from both the USA and China (Fig. 2c), suggesting hybrids that combine ancestry of multiple lineages (see section "Hybridizations in Japanese and Chinese mildews").

**Effective population sizes and cross-coalescence are correlated with historical migration patterns.** To study the diversification history of wheat mildew populations, we investigated how the different populations diverged over time. We reconstructed the population size variation and time of population split (through cross-coalescent rate variation) with coalescent methods (MSMC2). Where available, we selected eight isolates per population that best reflected their diversity based on PCoA, (Supplementary Fig. 12). Additionally, we calculated the fixation index ($F_{ST}$) that can be used as an approximation of the extent of population differentiation[45]. In our coalescent analysis, *B.g. dicocci* started with the highest effective population size ($N_e$), followed by an early decrease in $N_e$ (event 2, Fig. 3a). In the cross-coalescence analysis, *B.g. dicocci* were first to start separating from all others 3000–5000 generations ago (event 2, Fig. 3b). Assuming one sexual generation per year, and a constant mutation rate, we propose that this reflects the introduction of hexaploid wheat ~8000 years ago[5,46,47], which could have effectively limited gene flow between *formae speciales* specialized on wild tetraploid and domesticated hexaploid wheat. Indeed, *B.g. dicocci* had also some of the highest $F_{ST}$, Dxy and distance of average differences (DAD) values (Supplementary Figs. 13, 14), which suggested that it is largely genetically isolated, despite the hosts of both *B.g. dicocci* and *B.g. tritici* (bread wheat and wild tetraploid wheat) originating in the Fertile Crescent and both *formae speciales* being potentially able to infect either host. This is in accordance with previous studies[5,6,21,44,48,49]. Nevertheless, F4 statistics indicated that there is gene flow between *B.g. dicocci* and *B.g. tritici* lineages (Supplementary Table 4 and Supplementary Note 2).

In contrast, the Eurasian *B.g. tritici* populations (EUR-RUSK-ISR-CHN) showed slower separation from each other over time,

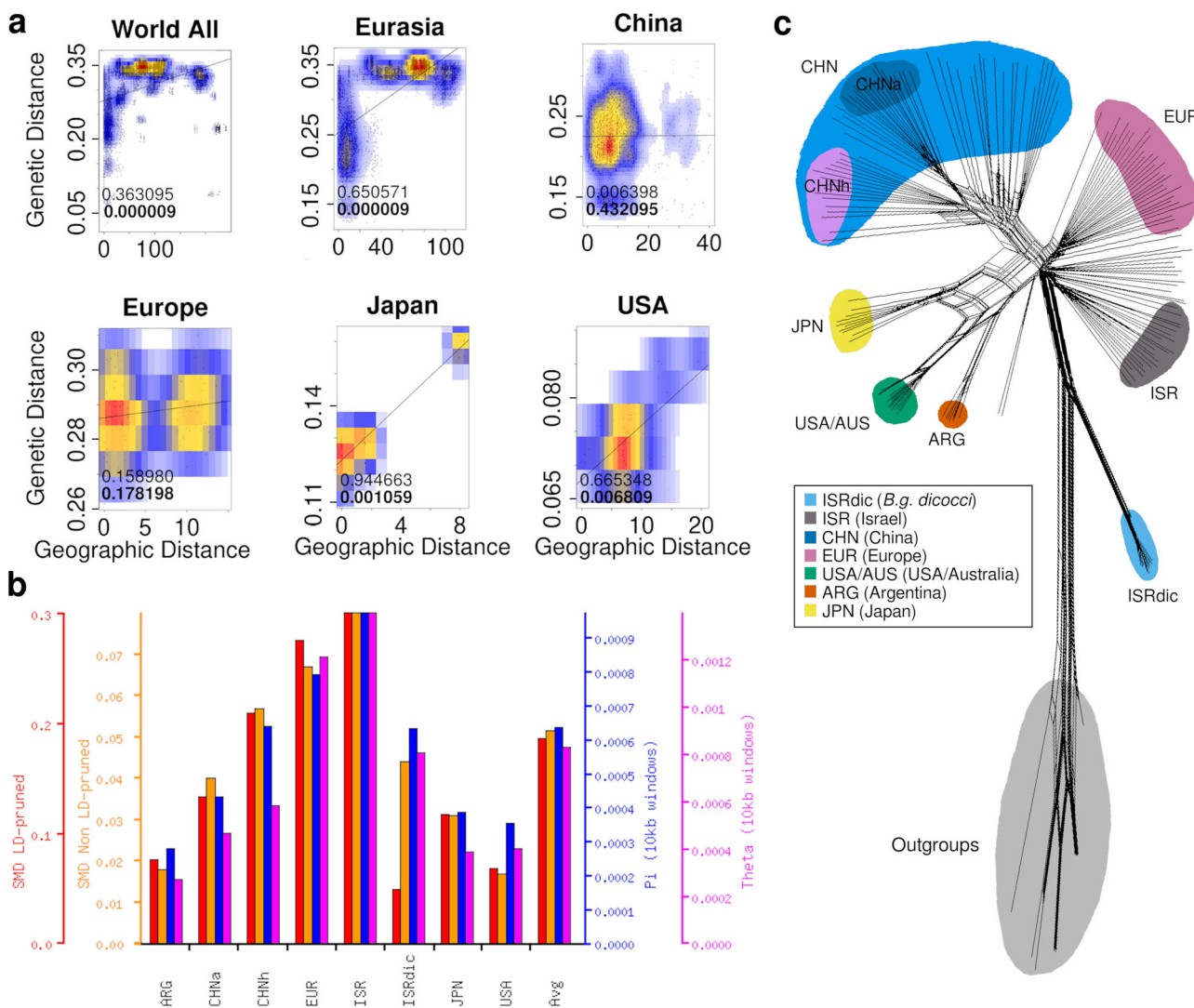

**Fig. 2 Population analyses of the wheat mildew populations. a** Correlation of the Geographic Distance (tens of kilometers by coordinates of the isolates) with the Genetic Distance (Simple Mismatch) using Kernel Density Estimation. In each plot, the Mantel test correlation and its *p* value (in bold) are shown. **b** Average differences within populations (ADW) diversity with simple mismatch LD-pruned SNPs (red bars), ADW diversity with simple mismatch unfiltered SNPs (orange bars), π genome-wide (average of 10 kb windows, blue bars), and Watterson's Theta genome-wide (average of 10 kb windows, pink bars). Note that *B.g. dicocci* shows low relative values of simple mismatch distances for LD-pruned SNPs, which could indicate the presence of multiple distinct lineages (see Supplementary Fig. 9 for details). **c** Phylogenetic network constructed with NeighborNet (SplitsTree software) using simple mismatch distances and a set of 2994 SNPs. The different main groups are colored. The isolates that are not colored belong to mildew populations of Turkey, Russia, Kazakhstan, and some isolates from Israel and China that do not cluster with the main groups (additional networks using different SNP datasets are shown in Supplementary Figs. 10 and 11).

which is in contrast to their rapid separation from USA and ARG populations, and the near-complete separation from the *B.g. dicocci* population (Fig. 3b). Here, $F_{ST}$ values of population pairs were lower compared to the American and *B.g. dicocci* populations, indicating that Eurasian populations were less differentiated from one another than from the American ones (USA-ARG; Supplementary Fig. 13). This could be explained by the long history of human migration and trade within Eurasia, as well as local wind dispersal. However, the distinct differentiation of the American mildews could also be the result of a founder event.

A notable recent feature, of all *B.g. tritici* populations, was a strong decrease in $N_e$ reaching similarly low levels approximately 50 to 100 generations ago (see methods). Although molecular dating of such recent events has to be taken with caution,

we propose that this could reflect a bottleneck created by modern breeding, starting in the middle of the 20th century, where monocultures of elite wheat cultivars replaced the older, much more diverse landraces. Widely used resistance genes, such as *Pm8*[50], and the use of modern fungicides and/or modern deep plowing of crop residues, may have contributed to selective pressure. After this decrease in $N_e$, most populations showed a rapid expansion. A similar recent decrease in $N_e$ was also found in the fungal pathogen *Rhynchosporium secalis*[51]. This could be the result of adaptation to modern crop cultivars or reductions in crop rotations.

An intriguing and different pattern exists between the mildew populations of the USA and Argentina. Firstly, both showed a drop in $N_e$ ~2000–5000 generations ago (event 3, Fig. 3a). As we argue in the next section, we propose that wheat mildews brought

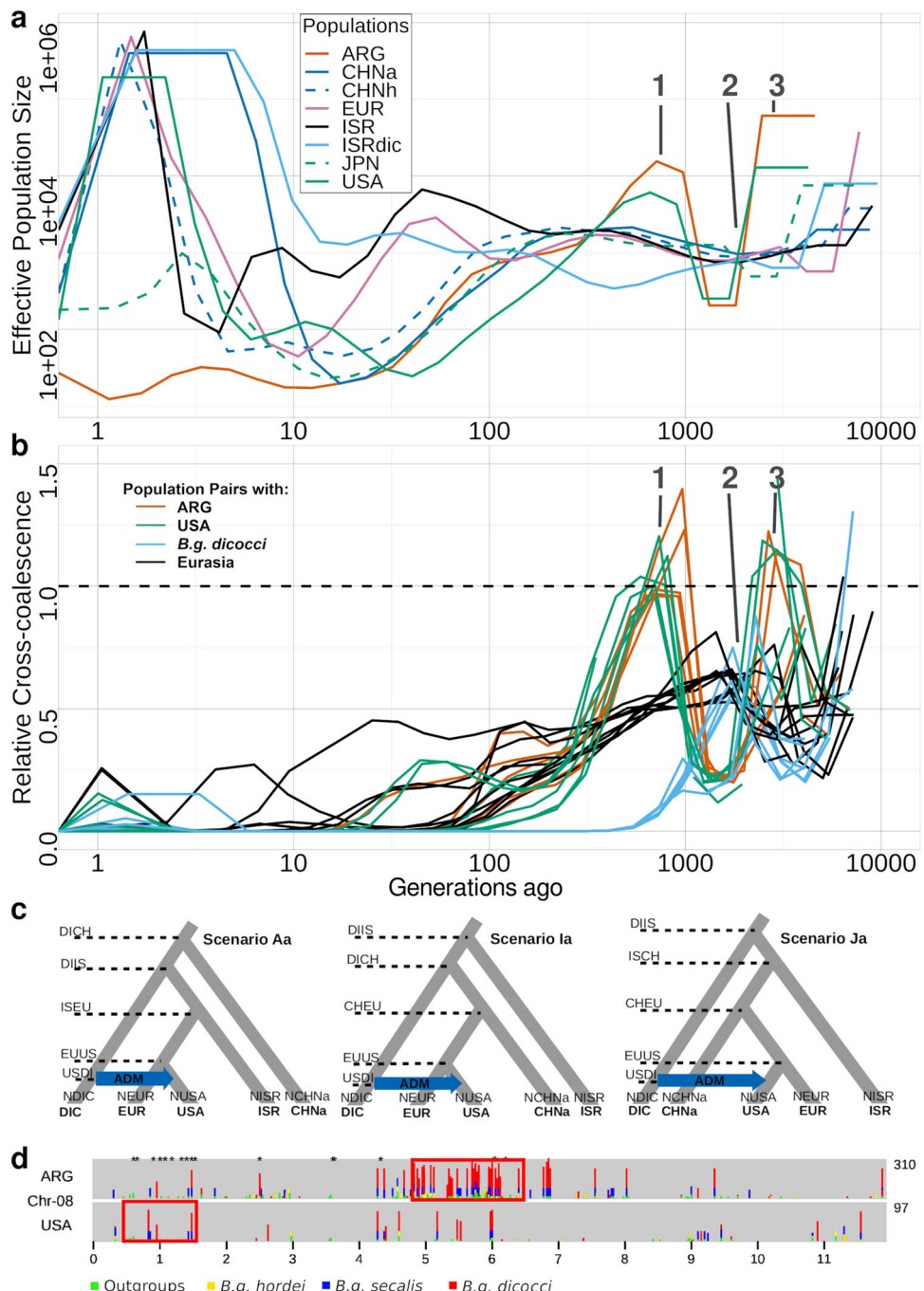

**Fig. 3 Effective population sizes and coalescence simulation. a** Effective Population size plot over time using the same mutation rate and eight of the most diverse isolates in populations. The dashed lines indicate the two recent hybrid populations. **b** Cross-coalescence analysis over time from the same dataset as in a. The blue line indicates a pair with *B.g. dicocci*, the green line indicates a pair with the USA population, the orange indicates a pair with the ARG mildew population, and the black line indicates a pair between the rest of the Eurasian populations. The dashed line indicates the threshold of one under which separation starts. The grey line with the number "1" refers to the late separation of the American populations from the rest (event 1) and the grey line with the number "2" refers to the approximate start of separation of the ISRdic from the other populations (event 2), while the number "3" grey line refers to the early separation of the American mildew populations from the rest (event 3). **c** Results of fastsimcoal2 analysis with the model schemes. The three best models are depicted (the rest are in Supplementary Fig. 17). The DeltaL and AIC values are depicted below the Scenario name in bold and non-bold letters, respectively. **d** Evidence for ancient hybridizations of mildews in North and South America. Mildew populations from USA and Argentina share numerous ancestral SNPs with distantly related (outgroup) grass mildews (colors indicate the number of outgroup isolates in which the SNP was found to be varied). The black asterisks on top indicate shared SNPs between the Argentinian and USA mildew populations. The red boxes indicate chromosomal windows that were under selection exclusively in the ARG or USA population.

to America hybridized with local grass mildews and the drop in $N_e$ could reflect that a small number of such mildew strains originally occupied the Americas. After populations (i.e., $N_e$) recovered, $N_e$ decreased again in both populations 200–500 generations ago (Fig. 3a), reaching minima ~100 generations ago. Additionally, American mildew populations had among the highest $F_{ST}$ values (Supplementary Fig. 13) and lowest sequence diversity. This could reflect the founder effect caused by the introduction of the pathogen to a new region, or a bottleneck due to specific environmental or agricultural conditions, such as widespread mechanization and use of a limited number of elite cultivars in the USA. Cross-coalescence indicated that mildews from the USA and Argentina derived from the Eurasian populations since they initially started to differentiate simultaneously in their early separation, more rapidly than the other populations (event 3, Fig. 3b,). Remarkably, both populations later showed twice a decrease in differentiation from the populations of Eurasia (event 1, Fig. 3b). We propose that the more recent differentiation and the decrease in $N_e$, reflects the introduction of powdery mildew to the Americas around 300–500 years ago, which geographically isolated them from European mildews. The earlier differentiation could indicate that mildews from the USA and Argentina experienced introgressions from distantly related mildews.

**Mildews from North and South America show evidence of ancient hybridizations.** The analyses described above consistently suggested that the mildews from the USA and Argentina may be hybrids between European and diverse, yet unknown, grass powdery mildews which were already present in America.

We tested this hypothesis through the identification of ancestral sequence variants and the unfolded site frequency spectrum (see methods). All populations showed the typical U-shaped distribution for derived allele frequency with two peaks at the extremes, indicating no substructure[52]. Here, alleles not occurring in *B.g. tritici* had the highest frequency ($x = 0$), followed by alleles found in the whole population (Supplementary Fig. 15). Furthermore, there was a peak in derived singletons very prominent in the European population, from which the American populations are presumably derived. In contrast, ancestral fourfold degenerate site SNPs showed that isolates from USA and Argentina do not have a strongly skewed distribution towards the ancestral allele, with very frequent alleles being enriched (Supplementary Fig. 16). This could indicate either a founder effect or genetic bottleneck leading to complete fixation of sequence variants in these populations.

To exclude alternative scenarios, 25 demographic models were tested for possible high gene flow from a different *forma specialis* into the USA population (Supplementary Fig. 17 and Supplementary Table 5), using coalescent simulation methods (fastsimcoal2) and the site frequency spectrum for the Israeli, European, USA, Chinese and *B.g. dicocci* powdery mildew populations. The most probable model proposes that the USA mildew population experienced gene flow from *B.g. dicocci* (Scenario Ia, Fig. 3c). At first glance, this finding was paradoxical. However, since we did not know the putative second parent, we proposed that the *B.g. dicocci* isolates share some ancestral sequence variants with that parent, which led to the fastsimcoal2 result (Supplementary Note 3).

If gene flow from a distant relative had indeed occurred, we expected to find diagnostic sequence variants common between the mildews from the USA and/or Argentina and *B.g. dicocci* or other *formae speciales*. Indeed, we identified 648 SNPs that occurred in isolates from the USA and in at least one distantly related mildew but were absent from populations from Europe. In Argentinian isolates, we found 1616 such SNPs, while 192 were found in both. These SNPs are spread unevenly across the genome (example in Fig. 3d and Supplementary Figs. 18, 19), with the central ~2.5 Mb of chromosome 8 being particularly enriched in isolates from Argentina. These SNPs were highly enriched in genomic windows that show signatures of selection (Fig. 3d and Supplementary Figs. 18, 19), suggesting that the proposed hybridizations played a role in mildew adaptation in America (Supplementary Note 4).

**Hybridizations in Japanese and Chinese mildews.** The above finding that mildews from USA and Argentina are likely ancient hybrids, inspired us to examine more closely the Japanese and Chinese mildews for which admixture analysis indicated the presence of multiple ancestries (see above, Fig. 1d). When mildews hybridize, the resulting genome is a mosaic of chromosomal segments from the two parents[21]. With time and repeated sexual recombination, these parental segments could be further fragmented. Additionally, back-crossing with one of the parents could reduce the genomic contribution of the other parent[21]. To test the hypothesis of hybridizations, we searched for chromosomal segments that could be assigned to either of the two presumed parental populations. Lineage-specific sequence variants were defined as those found exclusively in one of the parental populations (see methods). We found that the genomes of both the Japanese and CHNh isolates are comprised of large blocks that can be assigned to either of their putative parent populations (Fig. 4a–c and Supplementary Figs. 20, 21).

Assuming Japanese mildews were indeed hybrids, when did this hybridization happen? In the Japanese isolates, we could assign over 47,000 sequence variants to either of the two parents. All analyzed isolates had similar compositions, with about half of the SNP variants being found exclusively in mildews from the USA and half in isolates from China (Fig. 4a). Additionally, we identified between 1462 and 2131 haplotype breakpoints per isolate (i.e., swaps between groups of consecutive SNPs inherited from either parent). The admixture F3 statistic test of the type (pop1, pop2; JPN) resulted in Japanese mildews having the lowest value with USA and CHNa populations as parents (Supplementary Table 6 and Supplementary Note 5). However, the value was not negative, and, therefore, not significant. Thus, to further explore allele sharing between JPN and USA or/and CHNa, we employed the F4 statistics of the type (JPN; pop2, pop3; *B.g. poae*/outgroup). Indeed, we found the highest and most significant positive values when the USA or CHNa are used as pop3, indicating preferential allele sharing (Supplementary Table 7, for example, for pop2 = EUR, pop3 = CHNa, Z-score = 14.89). Japanese mildews being hybrids between USA and CHNa was further supported by the TreeMix analysis, indicating one recent migration from the USA/Australian clade to Japan (Fig. 4d and Supplementary Fig. 22). These data indicated that the sequenced Japanese mildews were the product of a very recent hybridization event, which probably occurred sometime in the 20th century (see next section).

The putative hybrids of the CHNh sub-population, which we identified based on Admixture and TreeMix analyses, showed similar characteristics, with over 18,000 sequence variants coming from either a CHNa or European parent (Fig. 4b, c). Here, parental sequence variants contributed between 40–60% to the individual isolates' genomes (Fig. 4b), which could indicate multiple hybridization events. These findings were also supported by the negative (and significant) values from admixture F3 tests of the type (pop1, pop2; CHNh), which strongly supported the CHNh sub-population being hybrids between CHNa and European isolates (Supplementary Table 8). The spatial overlap of the CHNh and the CHNa sub-populations may be explained by the fact that Chinese wheat landraces were still grown at the

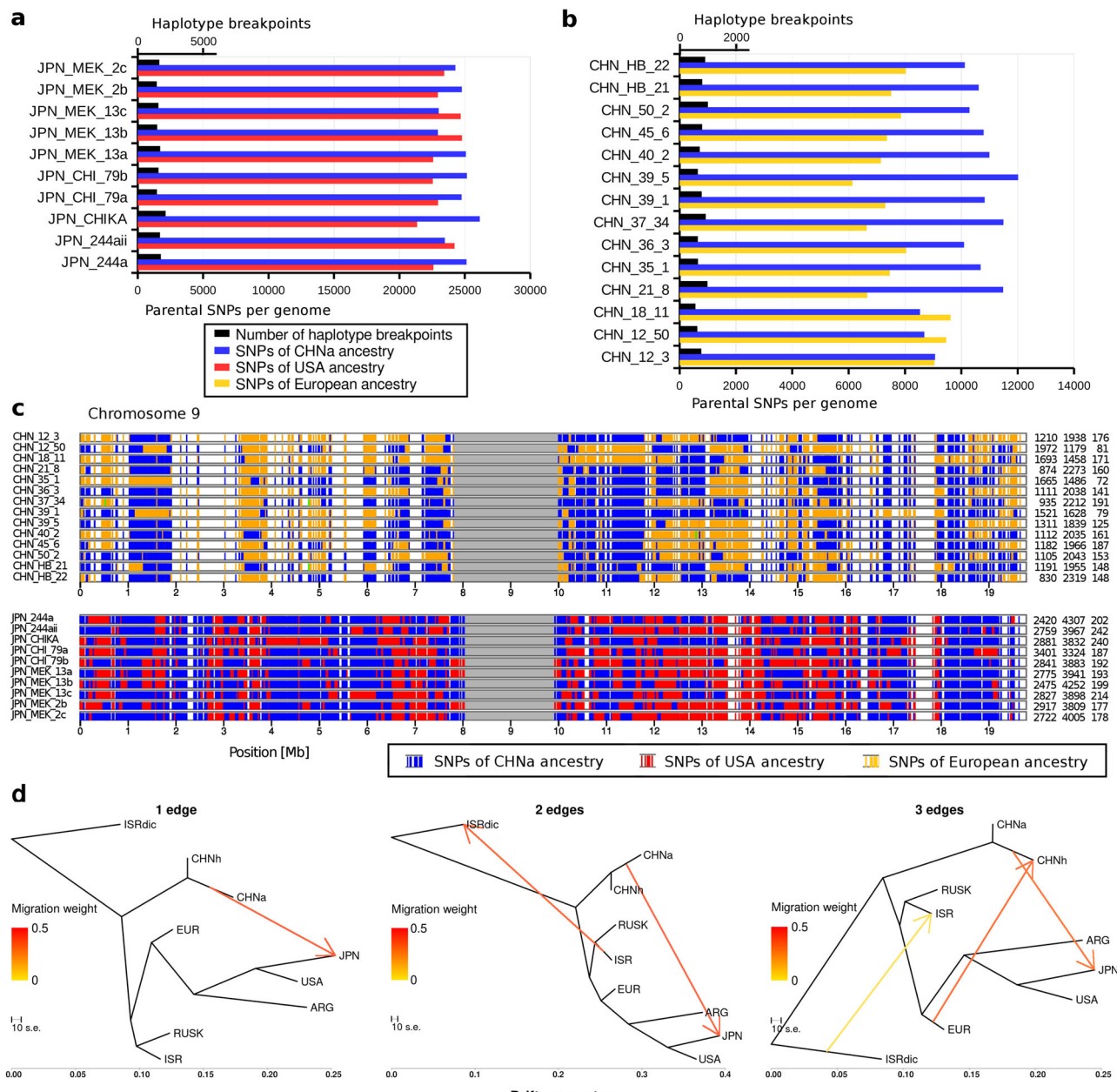

**Fig. 4 Genomic composition of young hybrid populations from Japan and China. a** Numbers of nucleotide variants and breakpoints between haploblocks (i.e., blocks of multiple consecutive SNPs inherited from a single parent), in putative hybrid populations from Japan. Mildew isolates from Japan seem to be hybrids between mildews from the USA and from China. In black are the numbers of breakpoints between chromosomal segments that come from either of the parents. In red and blue are the counts of identified nucleotide variants that could be assigned to either of the parents. **b** Analogous to (**a**). Mildews of the CHNh sub-population are likely hybrids between ancestral Chinese and European mildews. **c** Examples for the genomic mosaics that is caused by segments that come from different parental populations. Nucleotide variants are indicated by vertical bars. Colors identify the parental populations. **d** TreeMix analysis for one to three edges using *B.g. dicocci* as an outgroup. The red arrow indicates the migration event while the redness indicates its weight which approximates 50%. ISRdic refers to the *B.g. dicocci* population.

same time as modern European elite wheat cultivars entered Chinese breeding programs.

**A model for the worldwide spread of wheat powdery mildew.** In recent years, multiple fungal plant pathogen population genomic studies have become available[19,24,26,27,29–31,33,35,36,38,53], allowing comparisons with our findings. We found wheat powdery mildew had higher nucleotide diversity ($\pi$) than e.g., rice blast[36] and was less diverse than Dutch elm disease fungus[19] and wheat leaf rust[27]. However, $\pi$ was in the same order of magnitude

in all these studies. The observed differences could, for example, be due to fungal lifestyles, genomic repeat content, sample bias, or evolutionary history (bottlenecks, etc.). Additionally, most previous studies reported little geographical clustering of the populations. Instead, they showed that multiple different genetic lineages co-occur in the same geographical regions[19,24,27,29–31,36]. In contrast, we found a strong association between genetic and geographical distances between populations. Similar geographical clustering was found in *Z. tritici* and *P. nodorum*[26,33].

Although the spread of fungal plants pathogens has previously been associated with human activity, our study provides a more

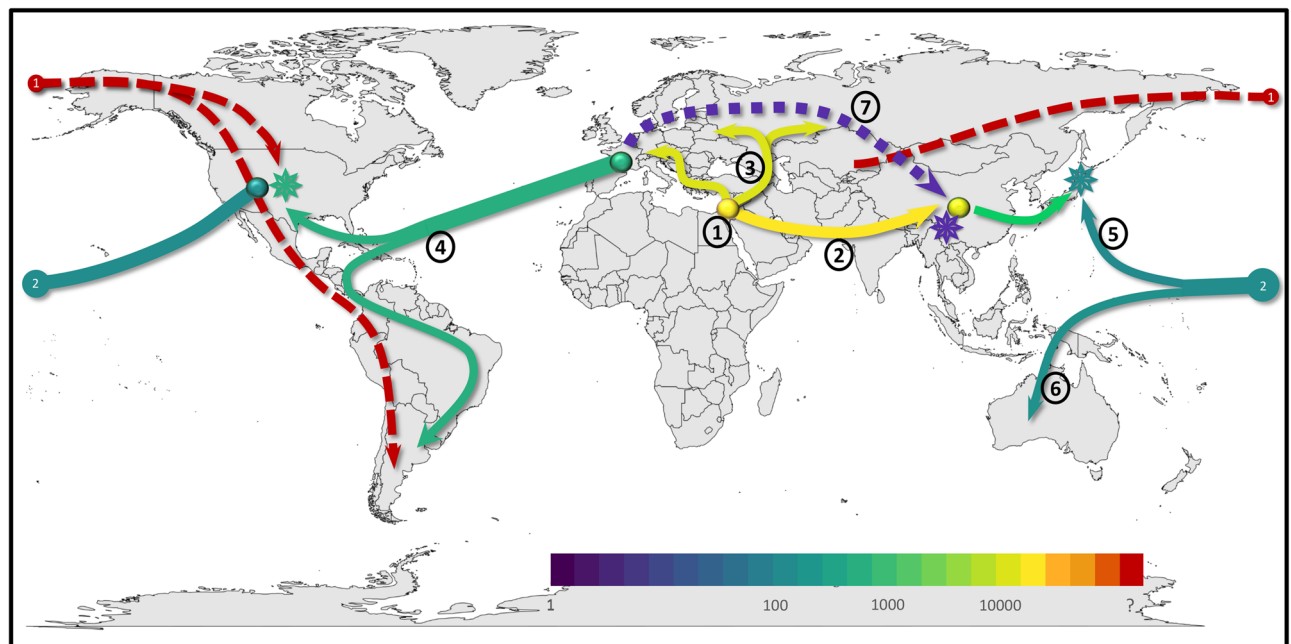

**Fig. 5 Worldwide migration patterns hypothesis over time and throughout the world.** Different colors represent the approximate time periods when the proposed events occurred. (**1**) Firstly, we have the Fertile Crescent being the origin of the pathogen. (**2**) Then, it splits, migrating to East Asia. (**3**) Migration of powdery mildew across all Eurasia. (**4**) During the colonization of America, Eurasian wheat powdery mildew was introduced to the Americas, which recombined with a distantly related powdery mildew, followed by isolation on this continent. (**5**–**6**) Later, the USA started exporting wheat to Japan and Australia, helping the pathogen migrate to both places. In Japan, because of the breeding of the American wheat lines with Japanese landraces, powdery mildew recombined between the East Asian and American populations and quickly got fixed because of adaptation to these new lines. (**7**) Finally, recent various levels of recombination between East Asian and European ancestry have been observed in the Chinese mildew population. The stars represent recombination events of populations that were fixed.

complete historical account, showing how wheat powdery mildew spread around the world, consistently following the introduction of its host to new areas (Fig. 5). Importantly, we found that oceans and mountain ranges effectively isolated mildew populations for centuries or even millennia. Powdery mildew of hexaploid wheat originated in the Fertile Crescent, when wheat cultivation started[5,6,46,47,54,55] (Fig. 5, step1). We propose that the emergence of hexaploid wheat led to the differentiation of *B.g. tritici* and *B.g. dicocci formae speciales* and their genetic isolation through host tracking[20]. Expansion of agriculture into Europe ~6500 years ago established distinct European mildews populations, which are closely related to those from the Fertile Crescent (Fig. 5, step3). We propose wheat landraces adapted to cooler climates drove the diversification of the European mildews. Additionally, our data strongly indicate an early introduction of mildew to China, possibly via trade on the Silk Road (Fig. 5, step2). Whether this coincided with early wheat cultivation in China ~4500 years ago[56] is unclear.

The introduction of wheat to America 300–500 years ago[7,57,58], also introduced European powdery mildew strains (Fig. 5, step4). Our data strongly indicate that mildews in North and South America soon thereafter hybridized with distantly related mildews. *Hordeum* species were shown to have colonized the Americas multiple times during the past four million years[59], and indigenous people may have cultivated certain crops, such as little barley[60]. It is, thus, conceivable, that mildews of wheat relatives were present when Europeans brought wheat to America. Indeed, a recent study identified various wild grass mildews in North America[61]. The high genetic similarity suggests that some mildew strains subsequently were brought from the USA to Australia, where wheat was introduced only in 1788[62]. (Fig. 5, step 5 and Supplementary Note 1).

Our analyses showed that more recently, powdery mildew was brought from the USA to Japan (Fig. 5, step 6), where it was again

hybridized with strains from East Asia. We propose that this is the result of intensifying trade between Japan and the USA following World War II. Wheat (and presumably wheat mildew) originally came to Japan from China over 2000 years ago[63–65]. During the past ~120 years of wheat breeding in Japan, germplasm was introduced mainly from North America and Europe[66]. We propose that recombination of Japanese landraces with North American cultivars also drove the hybridization of mildews from the USA and Japan, similar to the evolution of mildew of triticale[21]. Furthermore, we found that sometime in the past century, ancestral Chinese mildews hybridized with strains from Europe (Fig. 5, step7). This was presumably also driven by intensifying wheat imports into China from Europe[67]. This may be a general pattern as some previous studies also proposed hybridization as a possible factor in adaptation for fungal plant pathogens[19,30,31,36], with *Hymenoscyphus fraxineus* (ash dieback) being a particularly good example of how only two haplotypes that were likely spread by trade founded a new European pathogen population[34]. In summary, our data show how humans were the main vector for the spread of mildew throughout history, and how mildew rapidly and routinely adapted to new hosts through hybridization.

## Methods

**Fungal isolates collection**. A total of 224 sequenced isolates of *Blumeria graminis forma specialis tritici* used for at least one of the analyses. The isolates were introduced into the collection by either reviving the chasmothecia following the methods of Müller and colleagues[4], or by transporting and propagating conidia from leaves. Isolates that were acquired through chasmothecia were single-cell colony isolated to ensure that only one clonal individual was collected. Here, each isolate was propagated on 10-day-old freshly cut wheat leaves in Petri dishes. After four days, a single colony was picked with a toothpick and inoculated onto a new leaf. We repeated this twice to ensure one individual was isolated and then sub-cultured to obtain sufficient spores for sequencing. The dates of collection of the

isolates vary between 1990 and 2019 and the isolates are listed in Supplementary Data 1. For each country, isolates were collected from various fields and various cultivars from the species *Triticum aestivum* and *Triticum turgidum*. A list of the isolates, the country, the coordinates of collection, the year of collection, and the wheat species/subspecies that they were collected from is shown in Supplementary Data 1. As outgroups for some analyses, other *formae speciales* were used: *B.g. hordei*, *B.g. avenae*, *B.g. poae*, *B.g. lolii*, *B.g. secalis*, and *B.g. dactylidis*.

**DNA preparation and whole-genome sequencing**. In total, whole-genome sequencing was performed at different time points for 87 isolates. DNA extraction was based on chlorform- and CTAB based-protocol by Bourras and colleagues[68]. Illumina paired-end sequence reads of 150 bp read length and an insert size of ca. 500 bp were used to generate 1 to 6 Gb sequence data per isolate using either HiSeq 4000 or NovaSeq 6000 technologies. The remaining isolates were previously sequenced with the same DNA extraction protocol and their sequences were retrieved from the NCBI Short Read Archive (SRA). All Illumina sequence data is available from the SRA under project numbers shown in Supplementary Data 1.

**Read mapping and variant calling**. Raw Illumina reads were trimmed for adapter contamination and sequence quality. For trimming, the software Trimmomatic V0.38[69] was used with the following settings: illuminaclip=TRuSeq3-PE.fa:2:30:10, leading = 10, trailing = 10, slidingwindow = 5:10, minlen = 50. The reference genome *Blumeria graminis* f. sp. *tritici* CHE_96224 version 3.16 was used to align the reads by using the short-read aligner bwa mem v0.7.17-r1188[70] with the following settings: -M. We used samtools v1.9 for sorting and removing duplicate reads[71]. In order to mark PCR duplicates in the alignment (bam) files, we used the MarkDuplicates module of Picard tools version 2.18.25-SNAPSHOT. The average genome-wide coverage (calculated as the number of mapped reads multiplied by the average mapped read length and divided by the reference genome size) ranged between 10 and 60-fold for all the isolates with an approximate average of 45.46. Two isolates that had less than x20 average coverage were excluded from the analyses. We finally used "samtools index" to create indices for the bam files. The mapped reads were also further checked visually for their quality and GC% content, using the software qualimap2 and the multi-bamqc function[72]. GC content averaged 43.8% across isolates, with a standard deviation of around 0.9% (Supplementary Fig. 1). One isolate, JPN_MEK_13a, had more than six digits higher GC% than the average 43.8. This isolate contained small GC-rich contigs (mostly not mapped on the reference genome), possibly due to a sequencing artifact. It was eventually kept in the analyses.

The mapped reads were used to make variant calling files using GATK v.4.1.2.0[73–75]. We started by using the HaplotypeCaller with the options:–java-options "-Xmx4g", -ERC GVCF, -ploidy 1. Afterward, we used CombineGVCFs to combine the single gvcf files into one file and then, with GenotypeGVCFs we performed joint variant calls on the merged variant gvcf file using the option:–max-alternate-alleles 4. We used the GATK VatiantFiltration function to hard filter SNPs with quality thresholds recommended by GATK[74] and used with other plant pathogenic fungi[33,35]. The following thresholds were used: QUAL < 450, QD < 20.0, MQ < 30.0, $-2 <$ BaseQRankSum $< 2$, $-2 <$ MQRankSum $< 2$, $-2 <$ ReadPosRankSum $< 2$, FS $> 0.1$. Then, we filtered for a genotyping rate (>90%) and removed the InDels. The number of SNPs retained was 2,051,444. Finally, we made another dataset, where we kept only biallelic SNPs using vcftools:–max-alleles 2 (version 0.1.5)[76]. It was used for the site frequency spectrum (SFS) and fastsimcoal2 analyses. No SNPs were considered from the chromosome Bgt-chr-Un.

**De novo assemblies and mating types**. In order to compare specific genes and loci between the reference (CHE_96224) and different isolates, we created de novo assemblies for all the isolates. The trimmed reads were assembled using SPAdes genome assembler v3.12.0[77]. Then, ncbi+ blast v2.7.1+ was used in order to blast specific genes (e.g., mating type MAT genes)[78]. The mating type genes were either taken from the NCBI or identified by using mating type genes from closely related species. Then, the isolates were classified into two mating types. The Fisher's exact test was used on the entire dataset and on individual populations with more than eight isolates to check if there was a significant deviation from the expected 50/50 ratio of the two mating types. A chi-squared test was used to test for deviation from 50/50 in larger datasets (Supplementary Table 2). The tests were done with the R package stats (v.4.1.2).

**Calculation of genetic distances and identification of clonal isolates**. We used the SNPs from GATK to calculate the simple mismatch measure of dissimilarity, Bray–Curtis (Dice dissimilarity) distance, and individual Nei's genetic distance between all 224 isolates. Simple mismatch and Bray–Curtis distances were calculated with the vegan (v2.5-7) R package[79]. Nei's genetic distance[80] was calculated between individuals with the StAMPP package in R[81].

To identify clonal isolates (clone correction), a simple mismatch distance was used. Since, for clone correction, we used only LD-pruned SNPs (of which most are singletons), we assumed a close to a random distribution of SNPs. The threshold of minimum genetic distance was defined by using two isolates (ISR_30p and ISR_30w) that we knew are clonal and were separated for ~260 generations of

propagation. We then excluded one isolate each for pairs that had less than fifty times the simple mismatch distance between these two (which was about 0.004).

Additionally, we identified three isolates in the de novo assemblies with both mating types, possibly due to contamination. These were also excluded. The remaining 172 isolates that passed filtering were used with the rest of the analyses. We used the Pearson correlation coefficient in R with the cor() function to check for the correlation between the SNP quality (QUAL) and the alternative allele frequency (AF). They were highly correlated (Supplementary Fig. 2).

**Principal coordinate analysis (PCoA)**. After clone correction, the three genetic distances were re-calculated for the 172 mildew isolates. We used linkage disequilibrium (LD) pruned SNPs with a minor allele frequency (MAF) >0.05. In order to prune the LD sites, we used PLINK v1.9 with the parameters:–indep-pairwise 50 10 0.1. LD pruning and filtering resulted in 24,574 SNPs. The resulting vcf file was imported with the vcfR package (version 1.12.0) and transformed to a genlight object format in R. The PCoA analysis was performed using the glPca command from the adegenet R package version 2.1.3. We visualized the PCoA (along with other plots) with the ggplot2 package version 3.3.2. For the display, isolates were grouped according to country or region. We grouped the isolates from central Russia and Kazakhstan as middle Asia for this.

**Identification of ancestral alleles**. We used different outgroups to identify ancestral alleles in the SNPs for the demography, haploblocks, and the signatures of selection analyses. As *Blumeria graminis* belongs to a monophyletic group, the closest species with genomic sequences available are in the same family (*Erysiphaceae*) like *Erysiphe spp.* (e.g., *Erysiphe necator*). However, because it is a monophyletic group, only a few thousand common SNPs were left after the identification. Thus, we used instead the raw Illumina reads of six powdery mildew isolates of the same species, but from different *formae speciales* (three *B.g. hordei* isolates and one of the rest: *B.g. avenae*, *B.g. poae*, *B.g. lolii*). The pipeline that was used for these isolates was similar to the one for the *B.g. tritici* one. We only performed the GATK SNP calling and filtered for a genotyping rate of >50%, due to the low number of isolates, along with retaining only SNPs with no intra-specific polymorphism. Furthermore, the QUAL that was used for filtering was less than 50. After filtering, alleles that were identical in at least two of the three different *formae speciales*, were assigned as ancestral. 2,367,351 SNPs were assigned as ancestral (~34% of all the 6,905,274 identified filtered SNPs with the outgroups).

**Genome-wide statistics**. The statistics on the genome-wide coverage, minor allele frequency, and INDELs percentage were calculated as follows: We used bedtools coverage to calculate the mean coverage through the whole genome. We used vcftools to acquire the minor allele frequency of SNPs and INDELs for all individuals, along with the singleton information for the whole dataset. The site frequency spectrum was calculated for the different populations, in order to look at the distribution of the SNPs, using adegenet in R. Geographical collections had to have at least eight isolates to be considered as populations, as in previous studies in plants and animals[82,83].

**Genetic and geographic distance correlation analysis**. We tried to associate the genetic distance with the geographical distance between the isolates. Thus, for those 145 *B.g. tritici* isolates where exact sampling coordinates were available, we performed the Mantel test using all three genetic distances (simple mismatch, Bray–Curtis and Nei's genetic[80] distance). We used R and several dedicated packages[84]. The vcfR package version 1.12.0 was used to input the vcf files and change their format[85]. We then used the StAMPP package version 1.6.1 to perform Nei's genetic distance analysis[81]. To calculate the other two distances, we used the R package vegan using the genlight formatted files as input from vcfR. A matrix with the geographic distances was created using the dist() command of the stats package version 3.6.3. Then, we performed a Mantel test with a 100,000 repetitions using the mantel.randtest() command from the ade4 package version 1.7-15[86]. This test provides information on whether two matrices are correlated or not and, in our case, if the three distance matrices are correlated with the geographical distance one. We also used the kde2d command from the MASS package version 7.3-53 to calculate the two-dimensional kernel density estimate for all the correlated points in the genetic/geographic matrix[87]. We did all this on different levels: worldwide, continental level in Eurasia (excluding the Japanese mildew isolates due to their admixture) and finally, within populations for some powdery mildew populations in Asia. As the Mantel test does not have a strong power to detect correlation[71], we complemented it with the more sensitive distance-based Moran Eigenvector Maps (dbMEM) analyses on the same levels to try and find a correlation using the package memgene version 1.0.1[88,89].

**Population structure analyses**. We used different ways to investigate the population structure. We used the ADMIXTURE analysis and TreeMix. Admixture estimates maximum likelihood of the individual genetic ancestries from multiple SNPs datasets, while TreeMix uses a similar dataset to estimate historical relationships among populations, using a graph representation that allows both populations splits and migration events. We used the same SNPs set pruned with PLINK v1.9 as for the PCoA clustering. Firstly, we analysed the population

structure to get the possible ancestral populations with ADMIXTURE Version 1.3[90]. From the analysis, for ADMIXTURE the most probable number of ancestral populations was followed according to the lowest cross-validation error number, after testing for different number of Ks (1–15). Thus, the value of K = 7 was considered to be the most supported number of ancestral populations given the data. We tried the same analysis by subsampling the populations to check for consistency and the majority of the findings was similar to the previous ADMIXTURE analysis results (Supplementary Fig. 23). In the sub-sampled analysis we used 34 samples (Supplementary Table 9).

Moreover, to further look into the population structure, we looked at the recent migration patterns by using TreeMix v.1.13, using a varying number of edges from 0 to 15, taking *B.g. dicocci* as an outgroup, using the parameters: noss, k = 500 and bootstrapping[91].

**Estimating the *B.g. tritici* mutation rate.** Reference isolate CHE_96224 was sequenced originally in 2009 and sequenced again in 2018. During this time, the isolates were propagated ~5.2 times a year and, in between stored at 4 °C. This procedure roughly mirrors the number of generations (multiple asexual during mildew growth season) and one sexual generation during the year. The sequence contigs of the 2018 assembly were used for blastn searches (without low complexity filter) against the 2009 assembly. Because the assembly of highly repetitive regions was of poor quality in the 2009 assembly, only genes and 2000 bp flanking regions were considered.

The following steps were used to exclude sequencing errors, and false positives: (i) for sequence scaffolds with multiple near-identical hits (representing e.g. tandem duplications), only the top hit was used. (ii) SNPs that were closer than 100 bp to the end of sequences scaffolds were excluded because of generally lower sequence quality toward the end of contigs. Using these criteria, we estimated the mutation rate to be between $4 \times 10^{-7}$ and $5 \times 10^{-7}$ mutations per base pair per generation. We are aware that this mutation rate might be an overestimate, as deleterious mutations may be selected over time in a natural environment, leading to an overall lower evolutionary substitution rate over time.

**Phylogenetic networks.** For the construction of phylogenetic networks, we used simple mismatch individual distances, concatenated SNPs in coding sequences of genes, and a fourfold degenerate sites (4fdg) dataset. For this analysis we also included samples from other *formae speciales*, such as wild grasses powdery mildew. 4fdg SNPs were identified with an in-house script, which is stored in Github (https://github.com/caldetas/vcf_4fold_degenerate_dating_haploid). We kept 5039 4fdg SNP sites (excluding those on chromosome Unknown) to calculate genetic distances between isolates. We created a Kinship matrix according to these distances. For the calculation of the 4fdg distances, we used the Kimura-two-parameter (K2P) substitution model implemented in our in-house script (https://github.com/caldetas/vcf_4fold_degenerate_dating_haploid). Concatenated SNPs in coding sequences of genes were aligned with clustalw (version 2.1, obtained from Ubuntu repositories, ubuntu.com) using a gap creation penalty of 10 and a gap extension penalty of 0.2. The different datasets and distance matrices were used to check for the consistency of results. The SplitsTree4 (v4.17.1) software[92] was used to create the corresponding networks based on the three different datasets using the NeighborNet algorithm.

**Calculation of nucleotide diversity, genetic distances, fixation index, and Tajima's D.** Nucleotide diversity (π), Watterson's estimator (theta), and the number of singletons (i.e., SNPs found in only one isolate) were used to assess the overall genetic diversity of the different mildew populations, similarly to previous studies[21,93]. We estimated the whole-genome nucleotide diversity (π), in non-overlapping windows of 10 kb, assuming windows to be independent, for the different populations with the formula of Nei, implemented in the PopGenome R package[94], for all SNPs along with Watterson's theta (θ). The same analysis was repeated for eight randomly chosen isolates per population (Supplementary Table 10). They were also calculated for intergenic regions, genes, exons, introns, synonymous, and non-synonymous SNPs. Also, we used the simple mismatch distances to calculate the measure of average differences within populations (ADW) using LD-pruned as well as non-LD-pruned SNPs[95,96]. Here, LD-pruned, as well as non-LD-pruned SNPs, were used to check for consistency of results, because ADW values can be affected strongly by population structure.

We used SnpEff v4.3t[97] and SnpSift v4.3t[98] to annotate and predict the effect of all filtered SNPs. Then, from the output, we calculated the π and θ for the SNPs that had a low/medium/high effect annotated by SnpEff. Furthermore, we looked at the singletons count of SNPs across the genome. As there is a bias towards the reference, we were expecting an increase in the singletons of the European population were the reference is included (Supplementary Fig. 24). We then redid the analysis for a smaller dataset to check for similar patterns (Supplementary Data 2).

We also calculated distances between populations using the distance of average differences (DAD) with regard to the simple mismatch dissimilarity between individuals for the LD-pruned SNPset. Moreover, we calculated Dxy using the pixy (v1.2.5.beta1) software[99]. We used DAD in addition to Dxy, because DAD is a measure of "net" differences between populations.

To calculate the fixation index ($F_{ST}$) between mildew populations as an approximation of the extent of population differentiation, we used the R package PopGenome[94]. We calculated $F_{ST}$ for each chromosome separately and then estimated the weighted average (taking into account the chromosome size) without cut-off for minor allele frequency (MAF).

Tajima's D was calculated in 10 kb non-overlapping windows using the R package PopGenome[94] (Supplementary Fig. 25).

**F-statistics.** In order to clarify the possible shared ancestry and admixture between populations, we used the F3 and F4 statistics, calculated with the qp3pop and qpDstat packages in ADMIXTOOLS version 6.0[100].

The F-statistics tests measure allele frequency correlations between populations. In the case of F3, we can test whether a target population C is admixed between two parental populations (A and B). The F4 test can be used with four populations, when we have a non-admixed outgroup population, to test if there is any gene flow between the other two and a fourth population.

We used the F3 to indicate possible hybridization for the JPN population with the format (USA, CHNa, JPN) and for the CHNh population with the format (EUR, CHNa, CHNh). The results of these two analyses are shown separately in Supplementary Tables 6, 8. To explore the possibility of preferential gene flow between clades, we applied F4 statistics to sets of populations using LD-pruned SNPs with *B.g. poae* (BGPOA) as the outgroup. We designed the F4 comparisons as (ISRdic; pop1, pop2, outgroup [BGPOA]) with pop1 and pop2 being any pairs from our population's dataset. Here, we are looking for positive F4, indicating gene flow between ISRdic and other populations (the BABA configuration). We also used F4 with the model (JPN, pop1, pop2, BGPOA) in order to check for the population with the most shared drift with the JPN mildew population. In F4 statistics, we considered an absolute value of Z-score >3 significant.

**Demography.** To infer the demographic history of the populations, we used the Markovian coalescent MSMC2 (v2.1.1)[101]. Coalescence analyses use a backward-in-time algorithm, starting from the current generation and randomly building back the lineage of genomes. We investigated the effective population size of most geographic populations with more than eight isolates. Firstly, we created a SNPmappability mask using the reference genome v3.16. Then, we used vcftools to create SNP files from the bam files and mask files to cover the vcf files. After that, we diploidized the haploid vcf files and used generate_multihetsep.py to make the correct input format file for MSMC2. Then, we ran both the demography and afterward the cross-coalescence with 100 iterations and used eight isolates per population for eight wheat powdery mildew populations (Argentinian, European, Israeli, Japanese, *B.g. dicocci*, USA, CHNa, and CHNh), including only isolates without any recombination from the admixture analysis. In order to choose eight isolates, we created a SNP PCoA of each population and we manually chose eight samples with big distances from other samples (Supplementary Fig. 12). The most recent generations are not accurately reconstructed by these inference methods[102,103], therefore, they were excluded from interpretation.

We also used fastsimcoal2[104] (version 2.6.0.3) to simulate complex evolutionary models and estimate which ones would be more likely to be true. We were interested in testing the order of split events, along with the possible re-introduction of a EUR population ancestry to the USA or a split event from EUR to the USA. All the priors for these models/scenarios can be found in Supplementary Data 3. We used the unfolded SFS, with only derived alleles. For the models, we ran 100,000 simulations per replicate (-n) for 1000 replicates performing parameter estimation by max lhood from SFS values between iterations (-M) with 40 number of loops (ECM cycles) during lhood maximization (-L) and a 10 minimum number of loops for which the lhood is computed on both monomorphic and polymorphic sites (-l). The models had five of powdery mildew populations included: the European, the Israeli, the *B.g. dicocci*, the USA and the CHNa mildew populations. There were 25 scenarios tested, as seen in Supplementary Fig. 17. The best-supported scenarios are generally those with the highest probability (Supplementary Table 5). We also used the Akaike Information Criterion (AIC) as an estimator of prediction error, estimating the quality of the models.

**Identification of parental SNPs in hybrid populations.** To support the hypothesis that mildews from Japan and China were recent hybrids, we aimed to assign SNPs in the genomes of the putative hybrid populations (JPN and CHNh) to either of their two parent populations (i.e., CHNa and USA for JPN and CHNa and EUR for CHNh). SNPs were determined to belong to either of the two putative parents using the following criteria: A SNP in the putative hybrid has to occur in at least 70% of all isolates of the parent 1 population, but has to be completely absent from the parent 2 population, and vice versa. To address the possibility that an SNP might be simply absent from one parent due to undersampling, we ran the analysis multiple times by swapping a single isolate between parent 1 and parent 2. This reduced the number of SNPs that could be assigned to either parental population by 90–97%, depending on which two isolates were swapped. Our resampling approach indicates that the false positive rate of assigning an SNP to either of the two parents is 10% or less. Multiple consecutive sequence variants coming from one or the other parent were here referred to as "haploblocks".

To identify putative introgressions from distant relatives in mildews from the USA and ARG, we identified SNPs that occur in at least 85% (i.e., 7 of 8 and 14 o 16, respectively) of the USA and ARG isolates plus at least one of the ancestral lines (*B.g. lolii*, *B.g. dactylis*, *B.g. avenae*, *B.g. poae*, *B.g. hordei*, and *B.g. dicocci*), but are completely absent in the European mildew isolates.

**Scans for selection—detection of selective sweeps**. We performed scans for signatures of selection, in order to find genes that might be selected for in the different populations or are enriched where there are ancestral SNPs in USA and ARG. We included only SNPs that had an ancestral state, coming from the analysis with the ancestral alleles. For the detection of selective sweeps, we used a multiple signatures test (RAiSD v2.9) using these SNPs.

We used RAiSD, a software that calculates the mu value to identify candidate windows of selection[105]. We excluded centromeres from the calculations and we kept only the windows that lay above the 0.999 thresholds of probable selection. Then, we compared the statistics obtained for all the windows.

**Reporting summary**. Further information on research design is available in the Nature Research Reporting Summary linked to this article.

## Data availability

The whole-genome short-read sequence data of the 87 newly sequenced mildew isolates generated in this study have been deposited in the NCBI's Short Read Archive (SRA) database under the accession code BioProject PRJNA625429. All mildew isolates' genomic sequences used in this study can be found on SRA ncbi and the accession codes can be found in Supplementary Data 1 (PRJNA625429, SRP062198). The wheat powdery mildew isolates that have been sequenced here are available from the powdery mildew collection of Prof. Beat Keller (University of Zurich, bkeller@botinst.uzh.ch). Since individual mildew isolates may die over time, access to all isolates cannot be guaranteed indefinitely. Source data are provided with this paper.

## Code availability

The accompanying source code for the 4fdg site K2P distance calculations and the haploblocks analysis that we utilized in this study are available at (https://github.com/caldetas/vcf_4fold_degenrate_dating_haploid) and (https://github.com/wicker314/haplobloc) respectively.

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

## Acknowledgements

This work was supported by the University of Zurich Research Priority Program grant U-702-21-01. E.A.-I., C.B., and K.K.S. were supported by the University Research Priority Program of Evolution in Action of the University of Zurich, the SNSF Sinergia project "Out of Asia", and the NCCR Evolving Language, Swiss National Science Foundation Agreement #51NF40_180888. K.K.S. was also supported by the grant Kakenhi 21H05366.

T.B. and K.K.S. are supported by JST CREST number JPMJCR16O3. P.C.C. was supported by the Long-Term Research Program (Polish Ministry of Agriculture and Rural Development, years 2015–2020): Fundamental Research for Biological Progress and Preservation of Plant Genetic Resources as a Source of Innovation and Support for Sustainable Agriculture and National Food Security; task 3.2: Monitoring changes in pathogenicity of the cereal biotrophic pathogens. We would like to thank Toshiake Tameshige for helping us come into contact with people who collected samples and helping with the transportation of them. We would also like to thank Richard Oliver for helping us come into contact with the people who collected samples.

## Author contributions

A.G.S., T.W., and B.K. designed the project and interpreted the results. A.G.S. and T.W. wrote the manuscript. T.W., J.G., and A.G.S. wrote the software for some of the analyses. A.G.S. and H.Z. propagated the isolates in the collection. T.B., C.C., P.C.C., R.B.-D., A.D., S.R.E., K.H., M.H., J.S.-M., A.I.M., V.S., T.Y., and A.G.S. collected isolates. A.G.S. and H.Z. produced spores for DNA extraction. A.G.S., M.C.M., and S.B. extracted DNA for sequencing. A.G.S., T.W., B.K., K.K.S., B.A.M., C.B., and E.A.-I. contributed expertise in population genetics analyses. A.G.S. performed statistical and DNA-seq data analyses. A.G.S. and T.W. performed bioinformatics and population genetics analyses.

## Competing interests

The authors declare no competing interests.
