## [Peer Review File · Nature Communications]

Reviewers' Comments:

Reviewer #1:

Remarks to the Author:

The submitted manuscript of Sotiropoulos et al.: "Global genomic analyses of wheat powdery mildew reveal historical co-migration with humans" is manuscript with standard form, as required. English of the manuscript is good, however, manuscript suffers from lack of careful proofreading. Introduction is sound and well done. Results are new and well described besides definition of particular populations and some formal and format errors (details below). Some figures and their descriptions need improvement. Bioinformatics and statistical analyses are comprehensive and at high level.

Definition of populations from the Central/Mid Asian, West/East Asian is not clear from the map or it could be marked in the XLS file. Also, it is questionable to call "Population" a set of less than about ten isolates (authors have to be aware that they just scratched the surface of mildew variability, especially in Euroasia). This needs to be refined.

At line 172 is reference to Fig. 3b in regard to the FST, but the figure is not showing the indices and the referred conclusion is not obvious from the figure. Better would be referring to the Supplementary Table 3.

The occurrence of the Fig. 3 (line 173) is preceding the Fig. 2 (line 184).

In several cases there is missing dot after the "Fig" (lines 181, 213, 215, 217, 244, ...).

Sentence on the lines 256-257 needs to be rephrased. It is confusing. It seems that the second part is missing some component. I'm not sure what you wanted to express.

The result at lines 286-290 is nonsense. It is obvious that the CHNh have to have the EUR ancestry since you selected it that way. Also the EUR contribution is dependent on your threshold during the selection. You need to rethink how to describe the EUR influence to CHN mildew and how to discuss the spatial overlap with the CHNa and keeping the distinction.

Line 638 – it would be better to use "redundancies" as "any clones"

Line 645 – unnecessary "s" at the end of the line.

Line 798 – the sentence: "The best supported scenarios are those with the highest probability" is out of context, since all 25 scenario is provided, but the probabilities aren't.

The Discussion is well done, but it is a pity authors did not discuss deeper the divergence timing from the B.g. hordei since they have the largest data set so far.

The Fig. 1 has wrong labeling of the c and d sections.

Fig. 2 the asterisk indicating Chinese isolates with Fertile Crescent ancestry is not present (section b).

Fig. 3: There is abandoned "c" at line 1028. Line 1050 "package in r" should be "package in R".

Lines 1053 and 1054 are promising red and black bars in the sections b and c, but green and purple bars are provided.

Supplementary data:

Line 3. "Fig 1." should be "Fig. 1. "

Line 17 "snps" should be "SNPs"

Description of the Fig. 7. Is insufficient, it lacks a description what it means and what is difference between the sections.

Fig. 8. Line 84: authors declare that they analyzed for up to K=10, but the figure is providing cross validation up to K=15 (line 88).

Fig. 17. The description is insufficient. Abbreviations at y axis are not explained.

Fig. 18.: lines 264 and 266 the "nd" should be "and". Inconsistent markings of sections "a" should be "(a)". Line 268: "B.g. avenae" is redundant. Line 278: "in" is redundant.

Table 6: "significant results with $Z < 3$ is" but the lowest Z score is 5.8. ???

Line 585 In this case it is better to use "highlight" instead of "font".

Table 8 is surplus. Explanation above.

Reviewer #2:

Remarks to the Author:

The manuscript by Sotiropoulos et al. aims to reconstruct the population history of the wheat powdery mildew *Blumeria graminis*, a species that nowadays is a major pathogen of wheat throughout the world. The study conducts a wide variety of population genetic analyses on a diverse collection of 173 isolates from 13 countries and 4 continents, including 76 isolates that are new and generated by the authors of this study. From their analyses, the authors propose a model of the history of migration and evolution, starting in the fertile crescent 10,000 years ago and spreading throughout the world much more recently through trade. Understanding the population history of this fungal pathogen is an important and interesting topic and the authors have assembled a very nice dataset to tackle the question.

The major issue of the study is the poor writing and explanation of the study's analyses and results. There is a very large number of statistical analyses of the dataset but very little explanation as to what the questions are. Nearly all paragraphs of the Results section beginning by describing the method / software used; in contrast, the questions asked are never clear. In my mind, the results need to be rewritten, starting each section with the major question that is being asked and the tests / analyses the authors performed to address the question.

In several places, it is not at all clear where the numbers / results obtained are coming from. For example, the authors use molecular clock analyses to estimate the time of separation of different lineages. To do so, they have to rely on the substitution rate, which they infer, but never report its value. Thus, it is really hard to understand and evaluate the quality of their dating estimates. Later on, it appears that their inferences on hybridizations are based on patterns from SNPs, rather than from haplotypes, which would be much more informative and meaningful.

In summary, this manuscript reports a dataset that has the potential to provide significant insight into the evolution of a major fungal pathogen of wheat. However, the lack of clarity in the presentation of results and the authors' approach to report the results of way too many population genetic tests without any clear explanation or justification, makes this work hard to understand and evaluate.

Reviewer #3:

Remarks to the Author:

Evsey Kosman

Reviewer report

The manuscript "Global genomic analyses of wheat powdery mildew reveal historical co-migration with humans" is significant contribution to study of structure and historical spread and development of wheat pathogen *Blumeria graminis* f.sp. *tritici* all around the world. This research can serve a model for study of similar systems. Therefore, there is no doubt that this research must be published, and Nature Communications provides a good forum for that. Nevertheless, I think the manuscript is not ready for publication in its current form. I would suggest to the authors to consider the following major issues in order to improve the text by making it more properly structured and transparent for a general reader along with clarification and justification of several "technical" approaches used in data analyses.

1. The Results section includes many elements of discussion (some examples are highlighted and commented in the main text). On the other hand, the Discussion section (perfectly written) is very short and looks like Conclusions. Therefore, there is a need in restructuring the manuscript (to separate discussion from results). Another possible solution could be to rename the current section "Results" to "Results and Discussion", whereas the current section "Discussion" could be renamed "Conclusions".

2. Most readers do not familiar with details of numerous methods of data analysis used in this study. Therefore, it would be helpful (both from scientific and educational point of view) to briefly explain objectives of each method and possible outcomes. This might also allow for better understanding general logics of data analysis and whole research.

3. Three disparate distances between individuals (Nei genetic distance = $-\ln(1 - \text{simple mismatch dissimilarity})$, Nei and Li = Dice dissimilarity, and Euclidean) were used for different analyses in this study. This fact raises a question about general consistency of data analysis because these three metrics differ by their properties! Unfortunately, such confusion can be found in many studies... It is important and necessary to choose a proper distance and perform all relevant analyses with the selected metric.

There are also several minor points that need correction, clarification or additional explanation (some comments were made directly in the text - see attached file).

Nei genetic distance is commonly used for measuring differences between populations. Did you use it for individuals? This is formally possible, but I have never seen such application of the Nei distance.

It seems that clonality was measured by a threshold level of dissimilarity between isolates. This does not exclude a possibility of association between different SNP loci. The latter problem was solved with LD pruning. Nevertheless, I think this does not guarantee a random distribution of SNPs (alleles) among individuals within a population (given set of isolates) in a case of clonal or mixing type of reproduction.

Do you agree?

Fig. 2b: Where is the asterisk?

Line 641: 0.02 is the value of genetic distance, correct? If so, why %?

Line 645: "s" - what is this?

Lines 1052-1054, Figs. 3b, 3c: there are no black and red bars; the colors are different.

Lines 721-728, Phylogenies: What kind of trees was generated? This is absolutely necessary information in the Methods section.

Line 731: Nucleotide diversity (π) is simple average of all pairwise distances between individuals. It is relevant if all loci are assumed independent. Is this the case?

Lines 179-184: What distance do you mean, between populations (F_{st}) or between isolates? (I guess between isolates, but this should be mentioned)

Is this correct that SNP genotypes of all isolates were determined for a same number of loci? Or there were "missing (incomplete) data" for some isolates at least; in other words, "length" of SNP genotypes varies for different isolates.

REVIEWER COMMENTS

Reviewer #1 (Remarks to the Author):

Reviewer's comment 1.1

The submitted manuscript of Sotiropoulos et al.: "Global genomic analyses of wheat powdery mildew reveal historical co-migration with humans" is manuscript with standard form, as required. English of the manuscript is good, however, manuscript suffers from lack of careful proofreading.

Our response

We have carefully proof-read the paper and eliminated typos and other errors. All these small correction are highlighted.

Reviewer's comment 1.2

Introduction is sound and well done. Results are new and well described besides definition of particular populations and some formal and format errors (details below). Some figures and their descriptions need improvement. Bioinformatics and statistical analyses are comprehensive and at high level.

Our response

Formatting errors and figure descriptions were improved (see also our responses below).

Reviewer's comment 1.3

Definition of populations from the Central/Mid Asian, West/East Asian is not clear from the map or it could be marked in the XLS file.

Our response

We have now added the definition of these populations to the main text, instead of adding it to the XLS file, so that it is more easily accessible and clearer for the reader (Lines 162, 196-197).

Reviewer's comment 1.4

Also, it is questionable to call "Population" a set of less than about ten isolates (authors have to be aware that they just scratched the surface of mildew variability, especially in Euroasia). This needs to be refined.

Our response

There are, in fact, frequently cited publications (e.g. Nazareno et al., *Molecular Ecology Resources*, 2017, 17(6):1136-1147, Li et al., *Insects*, 2020, 11(5):290) that 8 individuals are sufficient for population analyses, and this threshold is used for many studies. We added a statement to the methods section referring to these papers and explaining better the definition of populations in Lines 496-497.

Reviewer's comment 1.5

At line 172 is reference to Fig. 3b in regard to the FST, but the figure is not showing the indices and the referred conclusion is not obvious from the figure. Better would be referring to the Supplementary Table 3.

Our response

Indeed, the Supplementary Table 3 contains the relevant information. We are now referring to it instead of Fig. 3b (Line 178).

Reviewer's comment 1.6

The occurrence of the Fig. 3 (line 173) is preceding the Fig. 2 (line 184).

Our response

Since we have removed the reference to Fig. 3 from the previous comment this has also been resolved.

Reviewer's comment 1.7

In several cases there is missing dot after the “Fig” (lines 181, 213, 215, 217, 244, ...).

Our response

The missing dots have been added.

Reviewer's comment 1.8

Sentence on the lines 256-257 needs to be rephrased. It is confusing. It seems that the second part is missing some component. I'm not sure what you wanted to express.

Our response

The phrase “Conversely, there is only limited allele sharing (smaller F4 values) between the European and the other mildew populations...” has been changed and split into two sentences to make the point clearer that limited allele sharing could be due to recent separation of European mildew or due to higher read mapping frequency on to the European reference genome (Lines 267-268).

Reviewer's comment 1.9

The result at lines 286-290 is nonsense. It is obvious that the CHNh have to have the EUR ancestry since you selected it that way. Also the EUR contribution is dependent on your threshold during the selection. You need to rethink how to describe the EUR influence to CHN mildew and how to discuss the spatial overlap with the CHNa and keeping the distinction.

Our response

The distinction between CHNa and CHNh genotypes was done based on the non-biased population structure analyses, which gave us the first indication that from all our different populations, the mildews from Europe seem to contribute part of their ancestry to some Chinese mildew isolates. Subsequently, we tested this hypothesis using F3 statistics, which clearly showed the EUR and CHNa populations to be the parents of the CHNh population. We chose our threshold of >25% for EUR ancestry, because very low percentages of ancestries should not be over-interpreted. We emphasize now more clearly that our selection of CHNa and EUR ancestry was based on the non-biased population structure analyses (Lines 297-298, 302-303).

The point of spatial overlap is important and was not addressed sufficiently by us. Spatially, they can overlap because Chinese wheat land races were still grown at the same time as modern European elite wheat cultivars entered Chinese breeding programs. Thus, host plants for both CHNa and CHNh isolates were grown at the same time on different fields. We have added this point to the main text (Lines 304-306).

Reviewer's comment 1.10

Line 638 – it would be better to use “redundancies” as “any clones”

Our response

The phrasing was changed (Line 476).

Reviewer's comment 1.11

Line 645 – unnecessary “s” at the end of the line.

Our response

The letter was removed (Line 485).

Reviewer's comment 1.12

Line 798 – the sentence: “The best supported scenarios are those with the highest probability” is out of context, since all 25 scenario is provided, but the probabilities aren't.

Our response

This was simply intended as a general comment on what is a more likely scenario. We have now rephrased the sentence and refer to Supplementary Table 9 where probabilities are shown (Line 659).

Reviewer's comment 1.13

The Discussion is well done, but it is a pity authors did not discuss deeper the divergence timing from the *B.g. hordei* since they have the largest data set so far.

Our response

The focus of this analysis was divergence time estimates of the main *B.g. tritici* and *B.g. dicocci* clades. For our analyses, we use *B.g. hordei* and other distantly related mildews as outgroups. The problem here is in *B.g. tritici* and *B.g. dicocci* practically all reads can be mapped, whereas *B.g. hordei* and *poae* etc. are so divergent that only a fraction of Illumina reads could be mapped. This selects for sequences that are highly conserved, resulting in a divergence time estimate that is too low. Divergence time estimates between such distantly related species have to be done using independent genome assemblies where entire orthologous genes can be aligned. Such an estimate was done in our previous study (Wicker et al., Nature Genetics, 2013, 45.9: 1092-1096). For these reasons, we would prefer not to discuss this in the main text, but we have added this information to the methods (Lines 578-584).

Reviewer's comment 1.14

The Fig. 1 has wrong labeling of the c and d sections.

Our response

The Fig. 1 has changed to include the sections "c" and "d" labels appropriately.

Reviewer's comment 1.15

Fig. 2 the asterisk indicating Chinese isolates with Fertile Crescent ancestry is not present (section b).

Our response

Asterisks have now been added to the two isolates from China that share ancestry with the Fertile Crescent mildews.

Reviewer's comment 1.16

Fig. 3: There is abandoned "c" at line 1028. Line 1050 "package in r" should be "package in R". Lines 1053 and 1054 are promising red and black bars in the sections b and c, but green and purple bars are provided.

Our response

The abandoned "c" was removed from the figure and the text was changed to "package in R". Finally, the names of the colours have been changed (Lines 943 and 944, in the figure legends).

Supplementary data:

Reviewer's minor comments

Line 3. "Fig 1." should be "Fig. 1. "

Line 17 "snps" should be "SNPs"

Our response

All changes were made as requested (legends of Supplementary Fig. 1 and Supplementary Fig. 3).

Reviewer's comment 1.17

Description of the Fig. 7. Is insufficient, it lacks a description what it means and what is difference between the sections.

Our response

We have added a general explanation of the method to the legend of Supplementary Fig. 7, as well as descriptions of the individual panels.

Reviewer's comment 1.18

Fig. 8. Line 84: authors declare that they analyzed for up to $K=10$, but the figure is providing cross validation up to $K=15$ (line 88).

Our response

In fact, we did include $K=11$ to $K=15$ in the analysis, but we did not include the Admixture plots of these high K values because they are not informative. However, in order to show that the Cross-Validation Error keeps increasing and that the minimum is clearly below $K=10$, we included this information in Supplementary Fig. 8 panel b. We have rephrased the second part of the Supplementary Fig. 8 legend to make it clearer.

Reviewer's comment 1.19

Fig. 17. The description is insufficient. Abbreviations at y axis are not explained.

Our response

We added explanations for the abbreviations to the legend of Supplementary Fig. 17.

Reviewer's comment 1.20

Fig. 18.: lines 264 and 266 the “nd” should be “and”. Inconsistent markings of sections “a” should be “(a)”. Line 268: “B.g. avenae” is redundant. Line 278: “in” is redundant.

Our response

The typo and redundant information were corrected and panel labelling has been made consistent in Supplementary Fig. 18.

Reviewer's comment 1.21

Table 6: “significant results with $Z < 3$ is” but the lowest Z score is 5.8. ???

Our response

As also mentioned in the text, significant results are considered to be the ones with a value of $Z < 3$. However, in our case there is no lower value than that of 5.8 since we do not have sequences of one of the actual parents of JPN isolates (a hypothetical diverged East Asian mildew). We have now changed the Supplementary Table 6 legend and explained this clearly.

Reviewer's comment 1.22

Line 585 In this case it is better to use “highlight” instead of “font”.

Our response

We have changed “font” to “highlight” in the legend of Supplementary Table 7.

Reviewer's comment 1.23

Table 8 is surplus. Explanation above.

Our response

The tables are the same analyses, but they are referring to different combinations of populations to verify that they are being recombined. We believe that they both should be included to make it clearer for the reader, since they are the same analyses but in different populations and with a different input. We make this clearer now in a short statement in the methods section (Lines 610-611).

Reviewer #2 (Remarks to the Author):

The manuscript by Sotiropoulos et al. aims to reconstruct the population history of the wheat powdery mildew *Blumeria graminis*, a species that nowadays is a major pathogen of wheat throughout the world. The study conducts a wide variety of population genetic analyses on a diverse collection of 173 isolates from 13 countries and 4 continents, including 76 isolates that are new and generated by the authors of this study. From their analyses, the authors propose a model of the history of migration and evolution, starting in the fertile crescent 10,000 years ago and spreading throughout the world much more recently through trade. Understanding the population history of this fungal pathogen is an important and interesting topic and the authors have assembled a very nice dataset to tackle the question.

Reviewer's comment 2.1

The major issue of the study is the poor writing and explanation of the study's analyses and results. There is a very large number of statistical analyses of the dataset but very little explanation as to what the questions are. Nearly all paragraphs of the Results section beginning by describing the method / software used; in contrast, the questions asked are never clear. In my mind, the results need to be rewritten, starting each section with the major question that is being asked and the tests / analyses the authors performed to address the question.

Our response

To address the reviewer's concern, we have added introductory sentences stating the biological questions and/or rationale for the analyses that follow, whenever one such statement was missing. Making the biological narrative and questions clear each time should now make it easier for readers to understand and follow the manuscript. All newly added introductory statements are highlighted.

Reviewer's comment 2.2

In several places, it is not at all clear where the numbers / results obtained are coming from. For example, the authors use molecular clock analyses to estimate the time of separation of different lineages. To do so, they have to rely on the substitution rate, which they infer, but never report its value. Thus, it is really hard to understand and evaluate the quality of their dating estimates.

Our response

This was an oversight. We have now included the nucleotide substitution rate in the methods (Lines 566-567).

Reviewer's comment 2.3

Later on, it appears that their inferences on hybridizations are based on patterns from SNPs, rather than from haplotypes, which would be much more informative and meaningful.

Our response

It is possible that the reviewer points to a problem of unclear definitions: we defined a haplotype as a group of sequence variants (i.e. SNPs) that are inherited together from a single parent. Thus, by using blocks of multiple consecutive SNPs we inferred a haploblock. We have added a brief definition to the main text (Lines 287-288), to the legend to Figure 4 (Lines 948-949), as well as a more detailed one to the methods section (Lines 674-675).

Reviewer's comment 2.4

In summary, this manuscript reports a dataset that has the potential to provide significant insight into the evolution of a major fungal pathogen of wheat. However, the lack of clarity in the presentation of results and the authors' approach to report the results of way too many population genetic tests without any clear explanation or justification, makes this work hard to understand and evaluate.

Our response

As described in our response to the reviewer's previous comment, we have added introductory

statements on biological questions to all paragraphs in the results and discussion section (see response to **Reviewer's comment 2.1**). Additionally, we included explanations in the Methods section explaining what the tests do and what the rationale was for using them for the following tests: Mantel test, admixture, treemix, F3 and F4 statistics (Lines 526-528, 539-542, 605-608).

Reviewer #3 (Remarks to the Author):

Evsey Kosman
Reviewer report

The manuscript "Global genomic analyses of wheat powdery mildew reveal historical co-migration with humans" is significant contribution to study of structure and historical spread and development of wheat pathogen *Blumeria graminis* f.sp. *tritici* all around the world. This research can serve a model for study of similar systems. Therefore, there is no doubt that this research must be published, and Nature Communications provides a good forum for that.

Reviewer's comment 3.1

Nevertheless, I think the manuscript is not ready for publication in its current form. I would suggest to the authors to consider the following major issues in order to improve the text by making it more properly structured and transparent for a general reader along with clarification and justification of several "technical" approaches used in data analyses.

Our response

A similar point was raised by reviewer 2 (see responses to **Reviewer's comments 2.1 and 2.4**). In order to more properly structure the manuscript, we have added introductory statements describing the biological questions or the justification why an analysis was done based on findings described earlier in the text (all these newly added introductory statements are highlighted).

We have addressed this issue by adding more explanation in the methods for the general reader (for Mantel test, admixture, treemix, F3 and F4 statistics analyses, lines 526-528, 539-542, 605-608), while also improving the Results text in general to make it clearer.

Reviewer's comment 3.2

1. The Results section includes many elements of discussion (some examples are highlighted and commented in the main text). On the other hand, the Discussion section (perfectly written) is very short and looks like Conclusions. Therefore, there is a need in restructuring the manuscript (to separate discussion from results). Another possible solution could be to rename the current section "Results" to "Results and Discussion", whereas the current section "Discussion" could be renamed "Conclusions".

Our response

As suggested by the reviewer, we renamed the "Results" into "Results and Discussion" and the "Discussion" into "Conclusions" (Lines 131 and 379).

Reviewer's comment 3.3

2. Most readers do not familiar with details of numerous methods of data analysis used in this study. Therefore, it would be helpful (both from scientific and educational point of view) to briefly explain objectives of each method and possible outcomes. This might also allow for better understanding general logics of data analysis and whole research.

Our response

As stated in the responses to **Reviewer's comments 2.1, 2.4 and 3.1**, we took several measures to guide the reader better through the manuscript and explain methods and rationales: (i) we added introductory statements describing the biological questions or the justification why an analysis was done where such statements were missing and (ii) we added detailed descriptions and goals of the analyses to the methods section.

Reviewer's comment 3.4

3. Three disparate distances between individuals (Nei genetic distance = $-\ln(1 - \text{simple mismatch dissimilarity})$), Nei and Li = Dice dissimilarity, and Euclidean) were used for different analyses in this study. This fact raises a question about general consistency of data analysis because these three

metrics differ by their properties! Unfortunately, such confusion can be found in many studies... It is important and necessary to choose a proper distance and perform all relevant analyses with the selected metric.

Our response

We have intentionally used different methods to assess distances between isolates as they are useful to address different questions: First, we used Euclidean distance for a PCA in order to cluster the individuals based on sequence variants and we did not use it for further analyses. In our opinion, a PCA is an important and intuitive data display for readers that are not experts in population genetics. PCA is frequently used to present a first overview of diversity (Walkowiak et al., *Nature*, 2020, 588.7837: 277-283).

Second, we used Nei's genetic distance to identify similarities between individuals and populations. In our opinion, it generally strengthens the results, as these methods complement and support each other. Additionally, we used Nei's genetic distance for some further downstream analyses of genetic distance (e.g. for the Mantel test and dbMEM analyses).

Third, we found nucleotide diversity as introduced by Nei and Li (as the average number of nucleotide pairwise differences in all possible pairs in the populations) was the most appropriate to assess diversity within populations.

In summary, we feel that the methods used by us complement each other and support a clear presentation of the biology. We hope that with these arguments we have convinced the reviewer that we have used the proper distances for the respective analyses.

There are also several minor points that need correction, clarification or additional explanation (some comments were made directly in the text - see attached file).

Reviewer's comment 3.5

Nei genetic distance is commonly used for measuring differences between populations. Did you use it for individuals? This is formally possible, but I have never seen such application of the Nei distance.

Our response

We have indeed used it for individuals, as is shown in Fig. 1d. Even though it is usually used to measure differences between populations, we used it for individual comparisons. There are indeed many different metrics for measuring differences between individuals and Nei's genetic is only one of them. In our case, we consider it formally legitimate, and it was used in the same way by the authors of the software (Pembleton et al., *Molecular ecology resources*, 2013, 13.5: 946-952) and in a similar way, for example, by Joly et al. (*Methods in Ecology and Evolution*, 2015, 6.8:938-948). We have added this clarification and these references to the methods section (Lines 512-513).

Reviewer's comment 3.6

It seems that clonality was measured by a threshold level of dissimilarity between isolates. This does not exclude a possibility of association between different SNP loci. The latter problem was solved with LD pruning. Nevertheless, I think this does not guarantee a random distribution of SNPs (alleles) among individuals within a population (given set of isolates) in a case of clonal or mixing type of reproduction.

Do you agree?

Our response

Yes, we agree that there is no guarantee for a completely random SNP distribution among individuals within a population. However, a close to random distribution is indicated by our data: the observations of the Site Frequency Spectrum (SFS) suggest that there is no substructure in the populations and that after the fixed SNPs, the singleton SNPs have the highest counts in all populations except ARG. (Supplementary Fig. 16). We have added a statement making this argument to the methods section (Line 476-477).

Reviewer's comment 3.7

Fig. 2b: Where is the asterisk?

Our response

The asterisks have been added in Fig. 2b. (See also **Reviewer's comment 1.14**)

Reviewer's comment 3.8

Line 641: 0.02 is the value of genetic distance, correct? If so, why %?

Our response

It is indeed the value of genetic distance. The mistakenly added “%” symbol has been removed. (Line 481).

Reviewer's comment 3.9

Line 645: “s” – what is this?

Our response

The wrong “s” has been removed. (Line 485) (See also **Reviewer's comment 1.10**).

Reviewer's comment 3.10

Lines 1052-1054, Figs. 3b, 3c: there are no black and red bars; the colors are different.

Our response

The color labels have been corrected in the legend and text. (See **Reviewer's comment 1.16**) (Lines 943-944, Legend).

Reviewer's comment 3.11

Lines 721-728, Phylogenies: What kind of trees was generated? This is absolutely necessary information in the Methods section.

Our response

The trees that were generated using the Kimura's two parameter (K2P) substitution model to find the distances between isolates using only fourfold degenerate sites. These distances were then used with hierarchical clustering to construct the tree. This information has been added in the Methods section (Line 574-576).

Reviewer's comment 3.12

Line 731: Nucleotide diversity (π) is simple average of all pairwise distances between individuals. It is relevant if all loci are assumed independent. Is this the case?

Our response

We calculated nucleotide diversity (π) in sliding windows of 10kb, genes etc., across the genome and calculated the simple average of all pairwise distances between individuals, in that way the individual values were assumed independent. For these analyses we focused either on specific regions of the genome (e.g. genes or the introns), or at the whole genome in sliding windows. We explain this now in the methods section (Line 590). This was handled the same in previous studies on plant pathogen and plant hybridizations (Menardo et al., *Nature Genetics*, 2016, 48: 201-205, Paape et al., *Nature communications*, 2018, 9.1:1-13.). We are now also citing these studies in the methods section (Line 589).

Reviewer's comment 3.13

Lines 179-184: What distance do you mean, between populations (F_{st}) or between isolates? (I guess between isolates, but this should be mentioned)

Our response

Indeed we refer to distances between isolates here and we are using Nei's genetic distance. We mention this now in the beginning of the paragraph (Line 184-185).

Reviewer's comment 3.14

Is this correct that SNP genotypes of all isolates were determined for a same number of loci? Or there were “missing (incomplete) data” for some isolates at least; in other words, “length” of SNP genotypes varies for different isolates.

Our response

Depending on the dataset used, there was different type of filtering. For LD-pruned SNPs and haploblocks, where the number of loci per isolate matters the most, we were more stringent, using only SNPs that are found in 99% of the isolates. For the other analyses, we used SNPs that were found in at least 90% of all isolates, as commonly used in other studies, e.g. in *Zymoseptoria tritici* (Hartmann et al., *Molecular ecology*, 2018, 27(12): 2725-2741). After checking, various minimum number of isolates having missing data, 10% percent keeps most of SNPs (92.02%) when using the outgroups, while if there are no missing data (0%) on a SNP, this drastically reduces the number of SNPs to less than a quarter (22.61%). That is why we used SNPs present in 90% or 99% depending on what resolution was needed for the analysis.

Reviewers' Comments:

Reviewer #1:

Remarks to the Author:

All comments and suggestions were well addressed.

Reviewer #3:

Remarks to the Author:

Evsey Kosman

Reviewer report – Revision 1

I already said in my report on the original version that manuscript "Global genomic analyses of wheat powdery mildew reveal historical co-migration with humans" is significant contribution to study of structure and historical spread and development of wheat pathogen *Blumeria graminis* f.sp. *tritici* all around the world, and Nature Communications provides a good forum for publication of this research.

The revised version of the manuscript looks much better, explanations were considerably improved, and modifications made allow for relatively easy understanding of logics and instruments used to reach the research objectives.

However, some additional clarifications and corrections are still need. The most important issue is using three or even four disparate distances between individuals (one or two Nei genetic distances, Nei and Li dissimilarity, and Euclidean distance) for different analyses. This point was raised in the previous review, but I think the authors didn't address properly this issue and don't agree with their response. Employing metrics with different properties leads to inconsistency of data analysis and may distort outcomes. To deal with this problem, two alternatives could be suggested. The first and preferable one is to choose a proper distance and perform all relevant analyses with the selected metric. Yet, in many cases such "proper distance" is unknown. Then, the second approach could be performing all analyses with several distance measures, so that only consonant results obtained with majority of metrics are considered valid.

I believe this study will serve a model for analyzing similar systems by different research groups. Therefore, it is absolutely necessary to demonstrate how to perform data analysis with the most possible consistency. Such methodological issues are also important for a general reader. I would be glad to assist the authors to implement some analyses in a more accurate way.

The following is the list of my comments and suggestions (according to the line numbers) that were also made directly in the text (see attached file).

141 What is "genotyping rate"? Please, explain briefly.

143 What is "minimum/maximum depth"? Please, explain briefly.

154-155 The idea of PCA is data visualization. The same can be obtained with multidimensional scaling (MDS). MDS works with an arbitrary distance matrix. In your case it would be more consistent to run MDS with Nei's genetic distance between isolates instead or in addition to PCA.

161 I think "correspondence" and "proximity" instead of "correlation" and "composition", respectively, are more suitable words here

168-169 I think this is "clustering based on the UPGMA dendrogram with regard to the Nei genetic distance".

If so, this should also be mentioned in Fig. 1d. The heatmap there is structured according to the dendrogram.

172-173 *F_{st}* does not reflect relationships between populations (see my comments in line 508) though it is wrongly used for that in numerous publications. Several papers were published with this regard during the last decade.

I would suggest to calculate pairwise distances between the populations on the basis of dissimilarities (genetic distance) between isolates.

184 replace "whether we can correlate" by "association"

195 What does this mean "the first eigenvector correlates"?

Is this a kind of slang?

228-230 Two questions:

1. What distance was used?

2 What kind of trees was generated?

251 See my comments in line 591

465-466 These three inequalities are wrongly written. Should be $-2 < \text{BaseQRankSum} < 2$ etc.

475, 480 Reference? Is this the same as Nei's genetic distance?

Nei distance considered in

Joly, S., Bryant, D. & Lockhart, P. J. (2015)

is not the Nei standard genetic distance. So, what distance was actually used?

476 This looks like a way to make clone correction. So, why don't to say this?

476-477 I don't really understand this inserted statement.

496 Anyway, I would call this either "set" or "collection" rather than "population"

499 Instead of PCA, more logically consistent approach could be Principle Coordinate Analysis (PCoA) or multidimensional scaling (MDS or NMDS) with regard to the Nei genetic distance.

508 Fst for individuals?

Fst is not distance, it is fixation index.

It does not generally reach maximum for two populations/individuals that do not share common alleles, which is counter intuitive.

Maximum of Fst is reached only for two populations/individuals with different FIXED (single) alleles. Therefore, Fst is not a proper metric for differentiation.

509 Not clear

511-513 Once again, Joly did not use Nei's standard genetic distance. What distance was actually used?

527 Nei genetic distance? If so, add Nei.

549 Supplementary Figure 20 itself does not appear, only the figure legend.

576 What kind of tree was generated?

591 Nei and Li; reference is missing

591-592 This seems a wrong interpretation of Nei and Li distance, which is actually the Dice dissimilarity.

To my knowledge, nucleotide diversity pi is usually measured with regard to the simple mismatch dissimilarity; then the interpretation seems correct.

I don't really understand the way of calculating Nei and Li (please, provide details), but it is possible that in this specific case values of Nei and Li distance and simple mismatch dissimilarity are equal.

679 Insert "in"

848-849 Journal name is absent

1041-1042 I think the following is a correct description of Figure 1d:

"Structured heatmap of relationships between isolates according to the clustering based on the UPGMA dendrogram with regard to the Nei genetic distance"

Fig. 1d This is a dendrogram (looks like the UPGMA one), not histogram.

What are the numbers (0.4, 1) along the vertical axes in the "color key"? I think they have to be removed.

According to the distance values shown in the "color key", the Nei distance used is not Nei's standard genetic distance as was stated several times in the main text! Please, check carefully this issue.

Fig. 3a Euclidean distance?

In the methods and legend, you mentioned Kimura-2 distance

Fig. 4 I would suggest to add here in the box a description of "black"; e.g.

"breakpoints assigned to the parents"

Reviewer #4:

Remarks to the Author:

In this manuscript the authors report on the invasion history of wheat powdery mildew, as inferred using a population genomics approach.

Main comments:

- Reviewer 2's comments have not been properly addressed, and the quality of the presentation of

research results could be improved. Having read this revised version of the manuscript, I can reiterate Reviewer 2's comments 2.1 and 2.4. The introductory sentences added by the authors often fail at stating clearly the biological questions or rationale for the analyses that follow. Some analyses addressing the same question are found in different paragraphs (e.g. population subdivision, colonization history), some results are mentioned before having been presented (e.g. diversity of the Chinese population), there is overlap between some figures, the vocabulary is at times awkward. More specific issues related to the presentation of research results are listed below in the "Other comments" section.

- The authors first infer the colonization history from phylogenetic trees, which is largely inappropriate since:

1/ The pathogen is sexually reproducing so relationships are not expected to be tree-like. Trees should be avoided and networks should be preferred.

2/ Trees and colonization history do not necessarily match. See doi: 10.1111/j.1365-294X.2010.04773.x

Approximate Bayesian computations (ABC) are only presented at the end of the paper, and mostly as a mean to infer the origin of American populations. It is not clear to me as to why there's no specific section investigating the colonization history – based on ABC only – after the analysis of population subdivision. Phylogenetic analyses (preferably using networks) could be used to introduce ABC, instead of being presented and used as a mean to infer the colonization history.

- What do we learn about invasive fungal pathogens? the broader implications of the present findings beyond the *Blumeria* case study could be clarified. The importance of this research could be discussed relative to other invasive fungal pathogens that have been equally well characterized. What novel insights have we gained on the emergence of fungal pathogens?

I think the research presented here would reach a wider audience if the paper was completely reorganized and the text thoroughly edited.

Other comments:

- Results section alternates between present and past tense. This should be past tense.

- L154: "To assess overall genomic diversity and to identify groups of similar isolates"

L172: "To answer the question of how related populations are to one another"

L204: "How are the different mildew populations structured genomically? To answer this question,..."

These three questions are largely similar, but still, they introduce different paragraphs.

- L115: "That study also proposed that wheat mildew itself is an ancient hybrid of mildews that grew on tetraploid or diploid wheat progenitors. Indeed, recombination may occur even between barley and wheat mildew formae speciales, which diverged millions of years ago."

The relationship between the two sentences is unclear to me.

Does it mean that because hybridization occurs between barley and wheat mildews, it is legitimate to hypothesize that wheat mildew is itself an hybrid? Please rephrase so that the reader does not have to wonder.

- L121: "Demography, diversity and/or selection was studied on global samples of plant pathogenic fungi using population genomics approaches e.g. *Zymoseptoria tritici*, *Rhynchosporium commune* and *Parastagonospora nodorum*."

These are studies that address questions that are quite different from the questions addressed here. Rice blast, wheat stem rust, boxwood blight, chestnut blight, dutch elm disease would be a much better examples.

- L124: "there are, to our knowledge, no genome-wide analyses on worldwide samples of obligate plant biotrophs."

Priority statements are always risky.

See DOI: 10.1111/mpp.12657

Or see <https://pubmed.ncbi.nlm.nih.gov/34544127/>
Probably others.

- L128: (ii) How diverse are mildew populations?

This doesn't sound like a research question but as a mean to answer a question. The level of diversity is not interesting per se; it's what it says about the population biology and evolutionary history of the pathogen that is interesting.

BTW, having read the whole manuscript, I'm unable to say whether mildew populations are more diverse than other pathogens, because the authors do not discuss this point.

- L133: It's not the worldwide diversity that you wanted to cover, but the species range. The worldwide diversity was unknown before this study, so it couldn't be known beforehand whether the proposed sampling would cover it.

- L138: Chi-square test for deviation from 50/50 expectation?

- L154: "To assess overall genomic diversity and to identify groups of similar isolates"
= To infer population subdivision, in population genetic terms.

- L155: principle component analysis -> principal component analysis

- L156: not clear what is a "Euclidean distance principle component analysis (PCA)". Is it a PCA carried out on a distance matrix? But the methods section suggest that this was actually done on a table of SNPs not a distance matrix. The proper approach to perform a multivariate analysis of a distance matrix is to use a Principal Coordinate Analysis, not a Principal Component Analysis.

- L161: "genetic composition of the isolates" is actually "population structure" in population genetic terms.

- L163: "The separate clustering of isolates from USA and Argentina was surprising, considering the presumably very short time of isolation since their arrival from Europe".

Is it the fact that USA and Argentina that do not cluster together that is surprising, or the fact that they do not cluster with European samples.?

None of the two possibilities is a surprise to me. Even if it's recent introduction, strong bottlenecks will lead to differentiation. Did you expect them to be in the European group?

- L167: "Interestingly, mildew isolates from Australia and the USA are indistinguishable, suggesting the pathogen reached Australia via the USA." What in the data, allow you to say that it's not the reverse. If there's nothing in the data, I wouldn't conclude about directionality. What's interesting here, I think, is that there's no differentiation despite thousands of kilometers, which suggest that introduction is very recent.

- L178: it does not "indicates", it "suggests". And the hypothesis of a lack of gene flow is testable, so why not test it?

- L154-L171: Is there anything that Nei's distance capture that could be missed by the PCA? So why use Nei's distance? The main conclusion is that (L169) "Nei's genetic distance also mainly separates isolates by geographic origin", but this is already obvious in the PCA.

Would be much more insightful to have an admixture plot here, instead of a figure that replicates the PCA.

- L172: Fst is not a measure of relatedness, it's a measure of differentiation. It measures differences in allele frequency, not the degree of consanguinity.

- L175: "closer" -> "less differentiated".

- L184: what's the hypothesis tested here? looking at this correlation is just a mean to look into a biological phenomenon. I guess it is dispersal that you are investigating here.

- L208: "very diverse individuals from China". But nucleotide diversity is presented later in the text, and the Chinese population doesn't look so diverse. And it is not *individuals* from China that are diverse, it is the *population* from China.

- L228: "To address the question of how the different populations diverged from one another". I think this is called the colonization history.

L234: "B.g. dicocci clustered clearly outside of B.g. tritici (Fig. 3a), although the hosts of both formae speciales (bread wheat and wild tetraploid wheat) most likely originated in the Fertile Crescent. This suggests that the region of origin of both formae speciales is also the Fertile Crescent."

I don't see how one can conclude about the geographic origin of the two lineages from such an analysis. It's over-interpreting data.

- Twelve occurrences of "Interestingly" in the whole manuscript. That's a lot.

- L251: "Looking at the overall genetic diversity, the population from Israel has the highest numbers of pi and theta (Fig. 3b and 3c, Supplementary Table 4), as well as high numbers of singletons in most isolates (Supplementary Fig. 11), supporting the hypothesis of the origin of B.g. tritici being the Fertile Crescent."

Singletons are characteristics of populations, not isolates.

I don't see why having a large number of singletons support a Middle east origin.

An excess of singletons suggests population expansion.

Can one can make a general prediction about the demography of populations in the center of origin?

It should be complex outcome of the status of wild crop relatives, the surface planted with modern domesticates, the use of fungicides or resistant varieties, and so on.

And why didn't the authors compute Tajima's D instead of counting singletons?

- L261: "Furthermore, we wanted to assess the levels of gene flow between populations." But F4 doesn't measure the level of gene flow. It just says whether there was gene flow or not. It's not quantitative.

- L321: how do the authors reconcile the observed excess of singletons in Israel, consistent with population expansion, with the recent bottleneck inferred using MSMC2?

- L637: "To check the demographic history of the populations"
Check?

REVIEWER COMMENTS

Reviewer #3 (Remarks to the Author, SEE ALSO ANNOTATED MANUSCRIPT, ATTACHED):

Evsey Kosman

Reviewer report – Revision 1

I already said in my report on the original version that manuscript “Global genomic analyses of wheat powdery mildew reveal historical co-migration with humans” is significant contribution to study of structure and historical spread and development of wheat pathogen *Blumeria graminis* f. sp. *tritici* all around the world, and Nature Communications provides a good forum for publication of this research.

The revised version of the manuscript looks much better, explanations were considerably improved, and modifications made allow for relatively easy understanding of logics and instruments used to reach the research objectives.

However, some additional clarifications and corrections are still need.

Reviewer’s Comment 3.1

The most important issue is using three or even four disparate distances between individuals (one or two Nei genetic distances, Nei and Li dissimilarity, and Euclidean distance) for different analyses. This point was raised in the previous review, but I think the authors didn’t address properly this issue and don’t agree with their response. Employing metrics with different properties leads to inconsistency of data analysis and may distort outcomes. To deal with this problem, two alternatives could be suggested. The first and preferable one is to choose a proper distance and perform all relevant analyses with the selected metric. Yet, in many cases such “proper distance” is unknown. Then, the second approach could be performing all analyses with several distance measures, so that only consonant results obtained with majority of metrics are considered valid.

I believe this study will serve a model for analyzing similar systems by different research groups. Therefore, it is absolutely necessary to demonstrate how to perform data analysis with the most possible consistency. Such methodological issues are also important for a general reader.

I would be glad to assist the authors to implement some analyses in a more accurate way.

Our response

This is a major comment which affected many analyses in the paper (albeit without changing the main findings and conclusions). Thus, our response here is lengthy and will be referred to in responses to subsequent comments:

After the discussion with the reviewer, we decided to use three types of distances between individuals on the same SNP sets, namely: i) simple mismatch distance, ii) Bray-Curtis (Dice dissimilarity) distance and iii) Nei’s genetic distance (Nei et al., 1972). This added robustness to the analysis and allowed us to evaluate results for congruency and consistency.

We used the three distances for the following analyses and/or data displays:

- 1) Principal Coordinate analysis (PCoA) plots
- 2) Identification of clonal isolates (“clone correction”)
- 3) Mantel test (1st test for association between genetic and geographic distances)
- 4) dbMEM (2nd test for association between genetic and geographic distances)

Overall, results for all three distances were very similar, indicating robustness of results. We gave priority to the simple mismatch distance in case of small incongruences between the three. For example, in the clone correction, this resulted in the removal of one additional isolate. Consequently, we re-did all calculations and analyses with this new dataset of 172

wheat mildew isolates (instead of the previous 173). Also the total numbers of polymorphic sites changed slightly.

The use of the three distances is now explicitly stated in the results (Lines 162-164) and described in the methods (Lines 488-491). Simple mismatch distance was used for main text and for displays in main figures, while results from the other distances are now shown in Supplementary Figures (5, 7 and 8) for comparison of results.

Our discussion with the reviewer also resulted in modifications on how we calculated diversity (see also comment 3.25):

To assess the within and between population diversity, we used multiple measures: first, we calculated π on the whole genome using non-overlapping windows of 10kb using all base positions (to allow comparisons with results of other studies, see comment 4.3). Then, average differences within population (ADW) on all LD-pruned SNPs and on non-filtered SNPs (excluding non-polymorphic bases) were calculated using simple mismatch distances of the individuals. This was done according to Kosman 2003 (*Plant Pathology* 52.5: 533-535) and Kosman 2014 (*European Journal of Plant Pathology* 138.3: 467-486), which we now also cite (Line 616).

This data is described in the main text (Lines 224-228) and in the new figure panel Fig. 2b (which also includes Watterson's Theta) and Supplementary Table 4. The populations showed the same relative levels of diversity in all four measures.

The only exception was *B.g. dicocci* (ISRdic), where simple mismatch distances of LD pruned SNPs were very low compared to the other SNP sets, which could indicate the presence of multiple distinct haplotypes. Indeed, PCoA analysis using non-LD pruned SNPs shows a split of the *B.g. dicocci* population into multiple subgroups. We only mention this briefly in the legend to Fig. 2, as we consider it a minor point, but show the additional PCoA data in the new Supplementary Fig. 9.

Reviewer's Comment

The following is the list of my comments and suggestions (according to the line numbers) that were also made directly in the text (see attached file).

Reviewer's Comment 3.2

141 What is "genotyping rate"? Please, explain briefly.

Our response

Genotyping rate is the proportion of individuals in the study for which the corresponding SNP information available (i.e. number of isolates that do not have missing data). We now explain this briefly in the main text (Line 148-149).

Reviewer's Comment 3.3

143 What is "minimum/maximum depth"? Please, explain briefly.

Our response

Minimum and maximum depth refer to the minimum and maximum number of reads, respectively, that map on a specific SNP. We are including this in the main text now (Line 151).

Reviewer's Comment 3.4

154-155 The idea of PCA is data visualization. The same can be obtained with multidimensional scaling (MDS). MDS works with an arbitrary distance matrix. In your case it would be more consistent to run MDS with Nei's genetic distance between isolates instead or in addition to PCA.

Our response

As suggested, we now use a Principal Coordinates Analysis (PCoA) MDS plot for the Main Fig. 1(b) and 1(c) and Supplementary Fig. 5. This point was also raised by reviewer 4 (see comment 4.15).

Reviewer's Comment 3.5

161 I think "correspondence" and "proximity" instead of "correlation" and "composition", respectively, are more suitable words here

Our response

We changed "genetic composition" to "genetic proximity" as suggested. However, for consistency of vocabulary we would prefer to replace "correspondence" with "association", as suggested by the reviewer in a later comment (3.8), Line 169-170.

Reviewer's Comment 3.6

168-169 I think this is "clustering based on the UPGMA dendrogram with regard to the Nei genetic distance".

If so, this should also be mentioned in Fig. 1d. The heatmap there is structured according to the dendrogram.

Our response

It was indeed based on an UPGMA dendrogram with regards to Nei's genetic distance. However, Reviewer 4 (comment 4.20) commented that the heat map displayed the same information as the PCA (now PCoA), and suggested removing the heat map. Additionally, reviewer 4 suggested to replace the main Fig. 1d panel with the admixture plot (comment 4.20). Accordingly, we now removed Nei's genetic distance heatmap.

Reviewer's Comment 3.7

172-173 F_{ST} does not reflect relationships between populations (see my comments in line 508) though it is wrongly used for that in numerous publications. Several papers were published with this regard during the last decade.

I would suggest to calculate pairwise distances between the populations on the basis of dissimilarities (genetic distance) between isolates.

Our response

The reviewer is correct, F_{ST} does not reflect relationships between populations, but is commonly used as a measure of population differentiation (Holsinger et al. 2009, Nat Rev Genet. 10: 639–650). We now clearly state in the text that we use F_{ST} only as an approximation of extent of population differentiation (Lines 252-253). Also, the wording in the methods was corrected (Lines 626-629, see also comments 3.18, 3.19 and 4.21 below). As suggested, we also calculated the pairwise distances of average differences between populations (DAD) on the basis of dissimilarities between isolates. We show the data as a network graph in Supplementary Fig. 14a) and refer to it in the text (Line 260).

Reviewer's Comment 3.8

184 replace "whether we can correlate" by "association"

Our response

We have changed the wording as suggested (Line 202) and throughout the following paragraph).

Reviewer's Comment 3.9

195 What does this mean "the first eigenvector correlates"?

Is this a kind of slang?

Our response

We have re-phrased the statement to and made the language more precise, saying that the eigenvectors separate (groups of) populations (Lines 214-217).

Reviewer's Comment 3.10

228-230 Two questions:

1. What distance was used?
- 2 What kind of trees was generated?

Our response

All trees were now replaced by phylogenetic networks, which was one of the main comments of Reviewer 4 (comment 4.2), because networks are more appropriate for recombining individuals (for details, see our response to comment 4.2).

To be consistent with our response to comment 3.1 and to assess congruence in the outcome of the networks, we used three distances to construct the networks:

- (i) The individual simple mismatch distances,
- (ii) The distances produced by the fourfold degenerate sites (4fdg) SNP differences using the Kimura-2-parameter model and
- (iii) Distances calculated based on an alignment of all polymorphic positions in coding sequences of genes.

The network derived from simple mismatch distances is shown in the main Fig. (now 2c), the others are presented in Supplementary Fig. 10 and Fig. 11, replacing the displays of trees (see also our response to comment 4.2).

Reviewer's Comment 3.11

251 See my comments in line 591

Our response

The reviewer refers to the use of nucleotide diversity (π). The detailed answer is given below (comment 3.25).

Reviewer's Comment 3.12

465-466 These three inequalities are wrongly written. Should be $-2 < \text{BaseQRankSum} < 2$ etc.

Our response

This was an oversight. Now the symbols have been corrected. (Lines 469-470).

Reviewer's Comment 3.13

475, 480 Reference? Is this the same as Nei's genetic distance?

Nei distance considered in Joly, S., Bryant, D. & Lockhart, P. J. (2015) is not the Nei standard genetic distance. So, what distance was actually used?

Our response

The citation of Joly et al. 2015 was removed. Instead, we cite the appropriate references for the StAMPP R package that was used to calculate Nei's genetic distance between individuals (Line 491). As described above, we now used simple mismatch distance for clone correction (see comment 3.1). The information is provided in the methods (Lines 492).

Reviewer's Comment 3.14

476 This looks like a way to make clone correction. So, why don't to say this?

Our response

We now use the term "clone correction" in the methods (Line 492).

Reviewer's Comment 3.15

476-477 I don't really understand this inserted statement.

Our response

We used this sentence to explain why we assume a close to random distribution of SNPs, but we see that it was described unnecessarily complicated. We have now simplified the explanation as follows:

"Since for clone correction we used only LD pruned SNPs (of which most are singletons), we assumed a close to random distribution of SNPs." Now in Lines 492-494.

Reviewer's Comment 3.16

496 Anyway, I would call this either "set" or "collection" rather than "population"

Our response

We have changed the sentence and now use "geographical collection" (Line 536).

Reviewer's Comment 3.17

499 Instead of PCA, more logically consistent approach could be Principle Coordinate Analysis (PCoA) or multidimensional scaling (MDS or NMDS) with regard to the Nei genetic distance.

Our response

As described in response to comment 3.1, we now use Nei's genetic distances, as well as simple mismatch and Bray-Curtis distances.

As suggested, we now use PCoA MDS, instead of the PCA, for all three distances. Now the Main Fig. 1 (b) and (c) panels show PCoA using the simple mismatch distances. Additionally, we show PCoAs using all three distances in Supplementary Fig. 5 for comparison. The clustering of populations was very similar for all three distances as well as compared with the initial PCA. This was also done in response to comments of Reviewer 4 (comment 4.15).

Reviewer's Comment 3.18

508 F_{ST} for individuals?

F_{ST} is not distance, it is fixation index.

It does not generally reach maximum for two populations/individuals that do not share common alleles, which is counter intuitive.

Maximum of F_{ST} is reached only for two populations/individuals with different FIXED (single) alleles. Therefore, F_{ST} is not a proper metric for differentiation.

Our response

We only used the F_{ST} for the populations (not for individuals), which we clarify now in the methods (Line 626). The inappropriate word "distance" was removed.

F_{ST} is usually used as measure for population differentiation, e.g. according to Holsinger et al. 2009 (Nat Rev Genet. 10: 639–650), but it is more correctly an approximation of extent of population differentiation, now used in Lines 252-253. These changes were also done in response to comments 3.7, 3.19 and 4.21.

With the proposed text changes, we therefore think F_{ST} is now used in the proper context and wording.

As for the maximum F_{ST} : we did not find F_{ST} values of 1 in any of the comparisons. Possibly, this was not described in enough detail.

To make F_{ST} results more easily accessible, we re-calculated F_{ST} for each chromosome separately and then calculated the weighted average (taking into account chromosome size). We considered this a simple and robust calculation, although final values differed only slightly from the previous ones which were based on genomic windows of 10 kb. F_{ST} values ranged from approximately 0.5 to 0.7 (for *B.g. dicocci*). We provide an overview of F_{ST} values as a heat map in Supplementary Fig. 13, instead of the previous Supplementary Table to

make differences better visible. We also provide a more detailed description of the F_{ST} values in the legend to Supplementary Fig 13.

Reviewer's Comment 3.19

509 Not clear

Our response

The explanation for F_{ST} calculation was revised (see comment 3.18 above).

Reviewer's Comment 3.20

511-513 Once again, Joly did not use Nei's standard genetic distance. What distance was actually used?

Our response

We used Nei's standard genetic distance (not the one used by Joly et al. 2015). The reference was removed (see also comment 3.13 above). Additionally, as mentioned above, the distances used now are: Nei's genetic distance, Bray-Curtis (Dice) dissimilarity and simple mismatch distances (see comment 3.1).

Reviewer's Comment 3.21

527 Nei genetic distance? If so, add Nei.

Our response

Yes, it is Nei's genetic distance. The reference was added (Line 542).

Reviewer's Comment 3.22

549 Supplementary Figure 20 itself does not appear, only the figure legend.

Our response

The figure indeed did not appear in the PDF file. We made sure that it does appear now in the provided PDF.

Reviewer's Comment 3.23

576 What kind of tree was generated?

Our response

All trees have now been replaced by phylogenetic networks as suggested by Reviewer 4 (see comment 4.2), since it is more appropriate with recombining individuals (for details, see our response to comment 4.2). As described above (comment 3.10), we constructed networks using three distances. The simple mismatch distance network is shown in Fig. 2c, the others are presented in Supplementary Fig. 10 and 11 for comparison (see also responses to comments 3.1. and 3.10 above).

Reviewer's Comment 3.24

591 Nei and Li; reference is missing

Our response

Nei and Li are not used anymore now (see comment 3.1). Instead we use Nei's diversity (Nei 1987, Molecular Evolutionary Genetics) and we are also citing the "PopGenome" R package that we used (Pfeifer et al. 2020, Package 'PopGenome'). Now in Line 611, see also next comment below.

Reviewer's Comment 3.25

591-592 This seems a wrong interpretation of Nei and Li distance, which is actually the Dice dissimilarity.

To my knowledge, nucleotide diversity π is usually measured with regard to the simple mismatch dissimilarity; then the interpretation seems correct.

I don't really understand the way of calculating Nei and Li (please, provide details), but it is possible that in this specific case values of Nei and Li distance and simple mismatch dissimilarity are equal.

Our response

We are now using the simple mismatch distance for individuals to calculate nucleotide diversity π with three different SNP sets, (i) LD pruned SNPs, (ii) unfiltered SNPs from across the genome and (iii) the average of π in 10kb windows across the genome. For the first two, we used the measure of average differences within populations (ADW), while for the last we used Nei's π calculated with the R package PopGenome (Lines 611 and 614-616, see also comments 3.1 and 3.25).

Reviewer's Comment 3.26

679 Insert "in"

Our response

The word "in" was inserted (Line 696).

Reviewer's Comment 3.27

848-849 Journal name is absent

Our response

The citation was removed in response to comments 3.13 and 3.20.

Reviewer's Comment 3.28

1041-1042 I think the following is a correct description of Figure 1d:

"Structured heatmap of relationships between isolates according to the clustering based on the UPGMA dendrogram with regard to the Nei genetic distance"

Our response

We have removed the Nei's genetic distance heatmaps now, in answer to the comment of Reviewer 4 (comment 4.20).

Reviewer's Comment 3.29

Fig. 1d This is a dendrogram (looks like the UPGMA one), not histogram.

What are the numbers (0.4, 1) along the vertical axes in the "colour key"? I think they have to be removed.

According to the distance values shown in the "colour key", the Nei distance used is not Nei's standard genetic distance as was stated several times in the main text! Please, check carefully this issue.

Our response

As described above, we have removed the Nei's genetic distance heatmaps now (see comments 3.28 and .20).

Reviewer's Comment 3.30

Fig. 3a Euclidean distance?

In the methods and legend, you mentioned Kimura-2 distance

Our response

Yes, Kimura-2 distances were used. However, the tree has now been replaced by a phylogenetic network as suggested by Reviewer's comment 4.2, since it is more appropriate with recombining individuals (see also response to comment 3.23).

Reviewer's Comment 3.31

Fig. 4 I would suggest to add here in the box a description of "black"; e.g.

"breakpoints assigned to the parents"

Our response

The figure panels 4a and 4c were not separated clearly enough, as the legend box belongs to 4c. The figure panels were re-arranged and an additional legend box was added for panels 4a and 4b.

Review #4 (Remarks to the Author):

In this manuscript the authors report on the invasion history of wheat powdery mildew, as inferred using a population genomics approach.

Main comments:

Reviewer's Comment 4.1

- Reviewer 2's comments have not been properly addressed, and the quality of the presentation of research results could be improved. Having read this revised version of the manuscript, I can re-iterate Reviewer 2's comments 2.1 and 2.4. The introductory sentences added by the authors often fail at stating clearly the biological questions or rationale for the analyses that follow. Some analyses addressing the same question are found in different paragraphs (e.g. population subdivision, colonization history), some results are mentioned before having been presented (e.g. diversity of the Chinese population), there is overlap between some figures, the vocabulary is at times awkward. More specific issues related to the presentation of research results are listed below in the "Other comments" section.

Our response

We have now revised and re-organized the text, also to address additional comments of reviewer 4 (comments 4.2, 4.4, 4.6 and 4.10).

The sections of the text were in part moved, in part merged to provide a narrative that is easier to follow. We decided to consider both technical aspects (going from general to more specific analyses), as well as the biological narrative (i.e. the spread of mildew from its region of origin through history across the world). The main topics are now ordered as follows:

- Summary information on genomes sequenced, mating types and number of sequence variants

- Identification and characterization of individual populations, and analysis of population structures.

- Association of genetic and geographical distances

- Inference of the region of origin

- Analysis of effective population sizes and coalescence

- Characterization of ancient hybridization of mildews in the Americas

- Characterization of recent hybridizations in Japanese and Chinese mildews

Furthermore, we added more detailed discussion of results from other studies (see comment 4.3).

Generally, we carefully revised the text with particular focus on the following:

- Introductory sentences have now been revised to more specifically describe the biological questions addressed in each respective section. All modified introductory statements are highlighted in the text.

- We avoided results being mentioned before being presented wherever feasible. If reference to later results was necessary, we added "see below".

- Finally, we carefully revised language and vocabulary.

Responses to more specific comments follow below.

Reviewer's Comment 4.2

- The authors first infer the colonization history from phylogenetic trees, which is largely inappropriate since:

1/ The pathogen is sexually reproducing so relationships are not expected to be tree-like. Trees should be avoided and networks should be preferred.

2/ Trees and colonization history do not necessarily match. See doi: 10.1111/j.1365-294X.2010.04773.x

Approximate Bayesian computations (ABC) are only presented at the end of the paper, and mostly as a mean to infer the origin of American populations. It is not clear to me as to why there's no specific section investigating the colonization history – based on ABC only - after the analysis of population subdivision. Phylogenetic analyses (preferably using networks) could be used to introduce ABC, instead of being presented and used as a mean to infer the colonization history.

Our response

We agree that networks are more appropriate than trees to represent closely related populations between which gene flow occurs. Consequently, we have replaced all phylogenetic trees with networks in the Main and Supplementary Figures. The networks were made with NeighbourNet (implemented in the SplitsTree software), which uses neighbor joining to construct the genealogic networks.

For the networks, we used simple mismatch distances between individuals, Kimura-2 parameter distances and networks constructed on multiple alignments of polymorphic sites in coding sequences of genes. Multiple distances were used in response to a main comment of Reviewer 3 (see response to comments 3.1).

The networks are more informative than the phylogenetic trees in the previous version of the manuscript, as they show strong grouping of most mildew populations and multiple connections of putative hybrid populations (e.g. USA, ARG and JPN) with their presumed parent populations. We replaced the discussion of phylogenetic trees with a discussion of the networks in the main text (Lines 234-244).

Additionally, a representative Network based on simple mismatch distances is now shown in Fig. 2c, while networks constructed with different SNP sets and distances are shown in Supplementary Fig. 10 and 11, to illustrate consistency of results. A section on the construction of the networks was also added to the methods (Lines 592-604).

As suggested, we also changed the order of the paragraphs, using network and population subdivision analyses to transition to ABC. With this, the narrative of the main text follows colonization history of wheat powdery mildew through history (see also our response to comment 4.1).

Reviewer's Comment 4.3

- What do we learn about invasive fungal pathogens? the broader implications of the present findings beyond the Blumeria case study could be clarified. The importance of this research could be discussed relative to other invasive fungal pathogens that have been equally well characterized. What novel insights have we gained on the emergence of fungal pathogens?

Our response

We have now added a section to the beginning of the Conclusions where we compare

findings of previous studies with ours. Here, we discuss specific aspects of individual studies, where results are directly comparable with ours (e.g. papers that calculated nucleotide diversity (π) in similar ways, or studied the association of genetic and geographical distances between populations). Lines 373-386 and 414-415.

Reviewer's Comment 4.4

I think the research presented here would reach a wider audience if the paper was completely reorganized and the text thoroughly edited.

Our response

We have now re-organized the Results and Conclusions section in order to have a clearer narrative (see also comment 4.1).

Additionally, we expanded the discussion of previous findings and how they compare to ours. We hope that these changes make it easier to follow the biological story and to reach a wider audience.

Other comments:

Reviewer's Comment 4.5

- Results section alternates between present and past tense. This should be past tense.

Our response

We have now changed all statements that refer to results of our analyses in the past tense. We only use present in cases where we refer to current ("real-life") situations. For example: "... Japanese mildews are recent hybrids between USA and East Asian mildews".

Reviewer's Comment 4.6

- L154: "To assess overall genomic diversity and to identify groups of similar isolates"

L172: "To answer the question of how related populations are to one another"

L204: "How are the different mildew populations structured genomically? To answer this question,..."

These three questions are largely similar, but still, they introduce different paragraphs.

Our response

The redundancy in these questions was removed since all introductory statements were revised to focus on the specific biological questions and hypotheses addressed in each section (see also comment 4.1 and 4.4).

Specific for this comment:

- Revised introductory statement on population subdivision (Line 162-163).

- The section dedicated to F_{ST} was removed. Instead, results of F_{ST} analysis are presented in individual sections complementing respective results, namely in the section on coalescence (Lines 259-260, 267-269 and 285-286). Additionally, we summarize results of F_{ST} analysis now in the new Supplementary Fig. 13 (see also response to comment 3.18).

- Revised introductory statement for the section on population admixture (Line 180-181).

Reviewer's Comment 4.7

- L115: "That study also proposed that wheat mildew itself is an ancient hybrid of mildews that grew on tetraploid or diploid wheat progenitors. Indeed, recombination may occur even between barley and wheat mildew formae speciales, which diverged millions of years ago."

The relationship between the two sentences is unclear to me.

Does it mean that because hybridization occurs between barley and wheat mildews, it is legitimate to hypothesize that wheat mildew is itself an hybrid? Please rephrase so that the reader does not have to wonder.

Our response

The idea was to emphasize that hybridization can occur even between very distantly related mildew lineages and even formae speciales. We re-phrased the statement accordingly. This is an important introductory sentence to our findings that mildews from America may be hybrids between wheat mildew and mildew from other grasses (Lines 116-118).

Reviewer's Comment 4.8

- L121: "Demography, diversity and/or selection was studied on global samples of plant pathogenic fungi using population genomics approaches e.g. *Zymoseptoria tritici*, *Rhynchosporium commune* and *Parastagonospora nodorum*."

These are studies that address questions that are quite different from the questions addressed here. Rice blast, wheat stem rust, boxwood blight, chestnut blight, dutch elm disease would be a much better examples.

Our response

We have added the suggested references and briefly summarize some of their main findings in the introduction (Lines 122-128). Additionally, we have added a section to the discussion where we compare our results that those of other studies (see comment 4.3.).

Reviewer's Comment 4.9

- L124: "there are, to our knowledge, no genome-wide analyses on worldwide samples of obligate plant biotrophs."

Priority statements are always risky.

See DOI: 10.1111/mpp.12657

Or see <https://pubmed.ncbi.nlm.nih.gov/34544127/>

Probably others.

Our response

We have toned down the language and made it more precise to explicitly refer to studies on whole-genome sequencing data (not e.g. reduced representation data such as genome-wide RADseq), and we cite now an example (Fellers et al. 2021, G3 11:9) (Lines 130-131).

Additionally, we provide a more extensive review of current literature, including studies that use other technologies such as RADseq or SSRs (Lines 125-127, see also comments 4.3. and 4.10).

Reviewer's Comment 4.10

- L128: (ii) How diverse are mildew populations?

This doesn't sound like a research question but as a mean to answer a question. The level of diversity is not interesting per se; it's what it says about the population biology and evolutionary history of the pathogen that is interesting.

BTW, having read the whole manuscript, I'm unable to say whether mildew populations are more diverse than other pathogens, because the authors do not discuss this point.

Our response

We re-phrased the biological question as follows: How is diversity of wheat powdery mildew populations associated with their evolutionary history? (Lines 134).

We are also comparing our values of whole-genome π in wheat mildews with available data from other studies in the conclusions (see also comment 4.3). There are indeed differences, but π is in the same order of magnitude in all the studies we cite. We also discuss possible explanations for the differences (Lines 374-378).

Reviewer's Comment 4.11

- L133: It's not the worldwide diversity that you wanted to cover, but the species range. The

worldwide diversity was unknown before this study, so it couldn't be known beforehand whether the proposed sampling would cover it.

Our response

As suggested, we replaced the term worldwide diversity with "species range" in the results section (Line 140).

Reviewer's Comment 4.12

- L138: Khi-square test for deviation from 50/50 expectation?

Our response

We performed Chi-squared test for the entire set of 172 isolates and Fisher's exact test on individual populations (with 8 or more isolates) to test if they deviate from the expected 50/50 ratio of mating types. We found no statistically significant deviation. This is now shown in the new Supplementary Table 3. We modified the statement in the Results section accordingly (Lines 144-145).

Reviewer's Comment 4.13

- L154: "To assess overall genomic diversity and to identify groups of similar isolates"
= To infer population subdivision, in population genetic terms.

Our response

We have changed the sentence to: "To infer population subdivision...". Line 162.

Reviewer's Comment 4.14

- L155: principle component analysis -> principal component analysis

Our response

The correct term "principal" is used now. However, we have now changed PCA to principal coordinate analysis (PCoA) in response to comments from both reviewers (comments 3.4., 3.17 and 4.15).

Reviewer's Comment 4.15

- L156: not clear what is a "Euclidean distance principle component analysis (PCA)". Is it a PCA carried out on a distance matrix? But the methods section suggest that this was actually done on a table of SNPs not a distance matrix. The proper approach to perform a multivariate analysis of a distance matrix is to use a Principal Coordinate Analysis, not a Principal Component Analysis.

Our response

The reviewer is correct. We are now using a Principal Coordinate Analysis instead of a PCA (also in response to comments 3.4 and 3.17). Additionally, multiple distances (simple mismatch, Bray-Curtis and Nei's genetic distance) were used in response to a comment of Reviewer 3 (see comment 3.1). Lines 162-165.

Reviewer's Comment 4.16

- L161: "genetic composition of the isolates" is actually "population structure" in population genetic terms.

Our response

Reviewer 3 suggested the use of the term "genetic proximity" (see comment 3.5), which we found to be more appropriate in this context (Line 170).

Reviewer's Comment 4.17

- L163: "The separate clustering of isolates from USA and Argentina was surprising, considering the presumably very short time of isolation since their arrival from Europe". Is it the fact that USA and Argentina that do not cluster together that is surprising, or the

fact that they do not cluster with European samples.?

None of the two possibilities is a surprise to me. Even if it's recent introduction, strong bottlenecks will lead to differentiation. Did you expect them to be in the European group?

Our response

We did, in fact, expect USA and Argentinian isolates to be closer together and closer to the European compared with the other populations, due to historical migration patterns (see also comment 4.18). Partially, the differentiation could indeed be due to bottlenecks, but our subsequent analyses also indicated gene flow from distant relatives (i.e. hybridization). We removed the term "surprising", replaced "founder effect" with "bottleneck" and changed the text to use more neutral language (Lines 172-174).

Reviewer's Comment 4.18

- L167: "Interestingly, mildew isolates from Australia and the USA are indistinguishable, suggesting the pathogen reached Australia via the USA." What in the data, allow you to say that it's not the reverse. If there's nothing in the data, I wouldn't conclude about directionality. What's interesting here, I think, is that there's no differentiation despite thousands of kilometers, which suggest that introduction is very recent.

Our response

The reviewer is right in that the genomic data itself does not imply directionality from USA to Australia, but it complements a strong historical case: if introduction had occurred from Europe through Australia to the USA, the ancient hybridization with a distant mildew strain would have had to occur in Australia, where wheat was introduced only in 1788, while it came to the USA already between 1600 and 1700. Furthermore, cultivation of *Hordeum* (little barley) is documented for Native Americans. Thus, a hybridization in Australia and subsequent migration to the USA would require that the new hybrids completely out-competed earlier mildew strains in the USA.

We now toned down the language in the results (Line 174-176) and modified the statement in the Conclusions section by also referring to historical accounts for the introduction of wheat to Australia in 1788 (Lines 402-404). To not expand the main text too much, we describe this historical argument in the new Supplementary Note 1.

Reviewer's Comment 4.19

- L178: it does not "indicates", it "suggests". And the hypothesis of a lack of gene flow is testable, so why not test it?

Our response

We tested for gene flow using F statistics. We saw allele sharing and gene flow that could be attributed to the recent or earlier geographical overlap. We have now added the information which shows that despite the differentiation of the *B.g. dicocci* population from Israel with the other populations, there is still gene flow (Lines 264-265).

Reviewer's Comment 4.20

- L154-L171: Is there anything that Nei's distance capture that could be missed by the PCA? So why use Nei's distance? The main conclusion is that (L169) "Nei's genetic distance also mainly separates isolates by geographic origin", but this is already obvious in the PCA. Would be much more insightful to have an admixture plot here, instead of a figure that replicates the PCA.

Our response

As suggested, we removed the Nei's genetic distance heatmap, moved the admixture plots in Fig. 1d and changed the text accordingly (Lines 162-179). Furthermore, the PCA derived

from the distances has been replaced by a Principal Coordinate Analysis (PCoA) (see also comments 3.4, 3.17 and 4.15).

Reviewer's Comment 4.21

- L172: F_{ST} is not a measure of relatedness, it's a measure of differentiation. It measures differences in allele frequency, not the degree of consanguinity.

Our response

F_{ST} is indeed used as a measure of population differentiation. However, Reviewer 3 suggested more cautious language, calling F_{ST} an approximation of extent of population differentiation (see also comments 3.7 and 3.18). We have now changed the main text accordingly (Lines 252-253), as well as the methods (Lines 626-629).

Reviewer's Comment 4.22

- L175: "closer" -> "less differentiated".

Our response

As suggested, we replaced "closer" with "less differentiated" (Line 268-269).

Reviewer's Comment 4.23

- L184: what's the hypothesis tested here? looking at this correlation is just a mean to look into a biological phenomenon. I guess it is dispersal that you are investigating here.

Our response

We now state that we had the hypothesis that wind dispersal could effectively spread mildew worldwide (Lines 203-204). This hypothesis was based on a previous study showing wind dispersal over hundreds of kilometers. Surprisingly, our data showed that wind dispersal is not sufficient for a worldwide spread.

Reviewer's Comment 4.24

- L208: "very diverse individuals from China". But nucleotide diversity is presented later in the text, and the Chinese population doesn't look so diverse. And it is not *individuals* from China that are diverse, it is the *population* from China.

Our response

This was an imprecise use of language. We were referring to how broad the Chinese data set was in terms of geographical and genetic distance (visualized in the PCoA). We were not referring to the nucleotide diversity. We now clarified the text (Lines 185-186).

Reviewer's Comment 4.25

- L228: "To address the question of how the different populations diverged from one another". I think this is called the colonization history.

Our response

We now replaced the trees with phylogenetic networks (see comment 4.2), which show more relationships between populations than population history (Lines 234-236). However, we integrated the term "colonization history" in the introductory sentence for the section on coalescence analyses (Lines 248-249).

Reviewer's Comment 4.26

L234: "B.g. dicocci clustered clearly outside of B.g. tritici (Fig. 3a), although the hosts of both formae speciales (bread wheat and wild tetraploid wheat) most likely originated in the Fertile Crescent. This suggests that the region of origin of both formae speciales is also the Fertile Crescent."

I don't see how one can conclude about the geographic origin of the two lineages from such an analysis. It's over-interpreting data.

Our response

The data for *B.g. tritici* clearly shows the highest diversity in the Fertile Crescent, which is generally seen to be a characteristic for the region of origin. However, we do not have comparable world-wide samples for *B.g. dicocci*. Thus, the reviewer is correct that we cannot conclude about the geographic origin of *B.g. dicocci*. We, therefore, narrowed the conclusion to be only about *B.g. tritici* (Lines 237-241).

Reviewer's Comment 4.27

- Twelve occurrences of "Interestingly" in the whole manuscript. That's a lot.

Our response

We have reduced the occurrence of the term "interestingly" to three times, and only used it to highlight particularly important or unexpected results.

Reviewer's Comment 4.28

- L251: "Looking at the overall genetic diversity, the population from Israel has the highest numbers of pi and theta (Fig. 3b and 3c, Supplementary Table 4), as well as high numbers of singletons in most isolates (Supplementary Fig. 11), supporting the hypothesis of the origin of *B.g. tritici* being the Fertile Crescent."

Singletons are characteristics of populations, not isolates.

I don't see why having a large number of singletons support a Middle east origin.

An excess of singletons suggests population expansion.

Can one can make a general prediction about the demography of populations in the center of origin?

It should be complex outcome of the status of wild crop relatives, the surface planted with modern domesticates, the use of fungicides or resistant varieties, and so on.

And why didn't the authors compute Tajima's D instead of counting singletons?

Our response

The reviewer is correct in that high singleton numbers do not imply region of origin. We have removed that statement. Instead, we build our argument on different measures of genetic diversity, as well as references to studies showing the origin of host (hexaploid wheat) being in that region (Line 223-225).

It is difficult to make general predictions about demographics in the region of origin with the available data, because a number of variables would have to be considered. Modern agriculture in the Fertile Crescent is very different from that in the past. For example, modern hexaploid wheat is grown across the Fertile Crescent nowadays (Royo et al. 2017, Mediterranean Identities: Environment, Society, Culture 16: 381-399), while wild tetraploid wheat (*Triticum dicocoides*) populations are usually small, in isolated niches distributed across the Fertile Crescent, and only rarely found in physical proximity to hexaploid wheat fields. Furthermore, the lineage of *Aegilops tauschii* that was donor of the wheat D genome is found exclusively in present-day Georgia (Gaurav et al. 2021, Nature biotechnology: 1-10). However, the status of wild wheat relatives may have fluctuated over time. What is more, due to the dry climate wheat powdery mildew epidemics are very rare in the Fertile Crescent. Thus, farmers do not usually use fungicides against powdery mildew nor mildew-resistant wheat lines. In contrast, climate was more humid more than 4,000 years ago in the northern Fertile Crescent (Pustovoytov et al., Quaternary Research, 67(3), 315-327). Considering all these factors, we would therefore prefer not to speculate too much about the role of wild crop relatives, fungicide use, surface planted etc.

As for the question about Tajima's D: We had originally calculated it, but concluded that the whole-genome Tajima's D values did not add substantial information, but merely supported

some findings (e.g. slight population expansion in the ISR population in recent times in MSMC2 analysis). However, since we did not want to over-interpret the MSMC2 results, we omitted it. To address the reviewer's comment, we now included the Tajima's D data for completeness in Supplementary Fig. 25 and refer to it in the methods (Lines 630-631). Additionally, we calculated between population diversity (D_{xy}), and the distance of average differences (DAD) values, as we considered these more informative measures and show the results also in Supplementary Fig. 14. We briefly mention both in the main text (Lines 259-260) and describe the procedure in the methods section (Lines 623-625).

Reviewer's Comment 4.29

- L261: "Furthermore, we wanted to assess the levels of gene flow between populations." But F4 doesn't measure the level of gene flow. It just says whether there was gene flow or not. It's not quantitative.

Our response

Indeed, F4 shows only if there is any gene flow. It also shows levels of allele sharing. We have now clarified this in the main text (Line 264-265). However, to save space we moved the summary section on gene flow and F4 statistics to the Supplementary Material (Supplementary Note 2).

Reviewer's Comment 4.30

- L321: how do the authors reconcile the observed excess of singletons in Israel, consistent with population expansion, with the recent bottleneck inferred using MSMC2?

Our response

It is important to note that the most recent generations are not accurately reconstructed by such inference methods, as has been stated in previous studies (e.g. Liu 2015, Patton 2019). Thus, the most recent generations should not be taken into consideration. We have now added this information in the methods section (Lines 663-665). We also now refer more specifically to the bottleneck found in the last 50-100 generations (Line 273). Furthermore, the observed excess of singletons could, in part, be explained by the fact that the powdery mildew population from Israel has overall the highest diversity. Additionally, gene flow from mildews that grow on wild or tetraploid relatives (e.g. *B.g. dicocci*) could also be a source of new singletons. Indeed, F4 statistics indicate gene flow from *B.g. dicocci* to *B.g. tritici*. We now describe this in the new Supplementary Note 2.

Reviewer's Comment 4.31

- L637: "To check the demographic history of the populations"
Check?

Our response

The word "check" was replaced with the more appropriate word "infer". Line 652.

Reviewers' Comments:

Reviewer #3:

Remarks to the Author:

The manuscript quality was considerably improved this time. The authors addressed all issues raised in my review, and the suggested changes were made in the revised version (data analysis, clarifications, terminology etc.). I believe the manuscript is ready for publication in its current form and recommend to accept it.

Reviewer #4:

Remarks to the Author:

I read Sotiropoulos et al.'s manuscript with great interest. Here are my suggestions for improving it:

1/ Title: Do the study really show that wheat mildew co-migrated with humans?

The abstract reads: "After it spread throughout Eurasia, colonization brought it to America, where it hybridized with unknown grass mildew species. Recent trade brought USA strains to Japan, and European strains to China."

I agree that is likely human migration that introduced the pathogen in the Americas, but regarding China or Japan, trade is not migration.

I understand that the authors wanted a catchy title, but it shouldn't be misleading.

2/ Research presentation: the reading of the results section remains hampered by the use of redundant statistics, by the use of too many "see below", or by hypotheses that are not well formulated.

There are too many "see below" in the results section (n=10 occurrences). Consequently, some sentences do not describe results from the paragraph in which they are placed, but results from a subsequent paragraph, which really hinders reading.

L178: I don't get why the authors wrote that "Chinese isolates (...) formed multiple subgroups with some showing similarity to European isolates". First, I don't see multiple subgroups, I see a gradient and two dots outside the main cloud. Second, China and Europe are clustering together along PCo2, not PCo1; is it sufficient to conclude that Chinese isolates "show some similarity to European isolates"?

L193-200: Admixture results should be described without making reference to directionality. For instance, there's no reason to write that "two Chinese isolates showed a high proportion of Israeli ancestry », as it suggests that Chinese isolates are – at least in part- from Israeli origin, while it is not what the analysis shows. The analysis shows that two Chinese isolates share ancestry with Israeli isolates, but it doesn't tell whether migration was from China to Israel or the reverse. Same comment about Japanese isolates.

L202-206: Wind is not the only factor that can generate an association between genetic distance and geographic distance, so the topic sentence of this paragraph sounds awkward. Humans, or their *Helicobacter pylori* pathogens, were not dispersed by wind, but still, there is a correlation between genetic distance and geographic distance in these organisms. doi:10.1038/nature05562

L222-228: Nucleotide diversity (π) is defined as the average number of nucleotide differences per site between pairs of DNA sequences within a population. What is the difference between π and the so called "average differences within population (ADW)" that the authors computed? If these two statistics are effectively different, what does "ADW" capture that π doesn't? Justification needed.

L226: one has to read the methods section to understand what "filtered" and "unfiltered" mean. And why do these statistics need to be computed with two different datasets? Justification needed.

L234: what is the point of constructing networks from several types of SNP sets? Do the authors expect differences in the historical signal captured by these different sets? Justification needed.

L248: "to study the colonization history of wheat mildew" could be the start of any paragraph.

L260: I'm aware that it was proposed by one of the reviewers, but I don't see the point of computing the "distance of average differences" other than to increase number of citations. I'm not a newbie in fungal population genetics but it's the first time I come across this statistic. I know that d_{xy} and F_{st} are complementary (doi: 10.1111/mec.12796); but what does DAD capture that d_{xy} or F_{st} don't? Justification needed.

L266: "In contrast, the Eurasian B.g. tritici populations (EUR-RUSK-ISR-CHN) showed more gradual separations from each other (Fig. 3b)." This suggest that you are comparing Eurasian populations to something? What does "gradual separation" mean? How can you tell separation is gradual on this plot?

L267 - 271: Alternatively, as you indicated elsewhere, it could just be that the differentiation between ARG and USA was inflated by a founder event?

L281: there was no wheat in the Americas 2,000-5,000 mildew generations ago (if generation time= 1 year).

Fig 2: too many digits in stats related to Mantel tests.

For the first row of Panel a plots, x-axis labels could be removed. For the second and third columns, y-axis labels could be removed.

Fig. 2b: histograms should be avoided. See

<https://journals.plos.org/plosbiology/article?id=10.1371/journal.pbio.1002128>

Figure 2b: what does "Average" stand for? The average across all window-specific values? The average of population specific averages?

Fig 2 caption: Capitalize Mantel https://en.wikipedia.org/wiki/Nathan_Mantel

Moreover, what does "results below it" mean? Is it the Mantel test statistic r ? I think the correct caption is Supp Fig 7's.

3/ Literature cited: the authors fail to cite and discuss important papers investigating the population genetics of introduced fungal pathogens.

For instance, McMullan et al. 10.1038/s41559-018-0548-9 is not cited.

Also, the authors cite papers that describe the population genetics of the chestnut blight pathogen in Britain or Tyrol instead of citing papers that investigated the population structure at larger scale, such as Stauber et al. 10.7554/eLife.56279 or Dutech et al. 10.1111/j.1365-294X.2012.05575.x

The authors cite LeBlanc et al. 2019 instead of LeBlanc et al. 2021 10.1094/PHYTO-06-20-0219-FI

L123 : Magnaporthae -> Magnaporthe. By the way, it would be unfair not to cite

<https://doi.org/10.1186/s12915-020-00818-z> and <https://doi.org/10.1128/mBio.01806-17> in addition to, or in place of, Zhong et al, since these papers used more representative sets of isolates, and more detailed analyses.

4/ Mutation/substitution rate:

Experimental evolution doesn't measure the substitution rate, but the mutation rate (L577). The substitution rate is estimated in calibrated phylogenies along longer time spans. Rates of molecular evolution measured over shorter time spans tend to be overestimated, as some sites will be

removed over time by negative selection. The rate presented here is more similar to a spontaneous mutation rate, than an evolutionary substitution rate.

The authors should be aware that several studies estimated substitutions rates in fungi, using tip dating in particular (e.g. *Cryptococcus gattii*, *Magnaporthe oryzae*, *Schizophyllum commune*). Estimates of the substitution rates are generally smaller than what is presented here (10⁻⁸/site/year, vs 10⁻⁷/site/generation here).

Would their conclusions regarding wheat - wheat mildew coevolution still hold with a different substitution rate? (L255-259)

This is really worth discussing.

5/ Other comments:

Fig1a: I can't spot the purple and grey dots, as they are too small, or colors are not appropriate. Do you really need to plot *T. turgidum* and *Triticum* spp?

L185: what is between parentheses should be a sentence. I don't see why it is between parentheses.

L199 delete ". K".

REVIEWER COMMENTS

We provide a PDF in which all changes are tracked (Manuscript_Main_Text_Changes_Tracked)
Line numbers in these responses to reviewers' comments refer to that document. Tracked are also the changes made in response to the editorial manuscript checklist.

Reviewer #3 (Remarks to the Author):

Reviewer's Comment 3.1

The manuscript quality was considerably improved this time. The authors addressed all issues raised in my review, and the suggested changes were made in the revised version (data analysis, clarifications, terminology etc.). I believe the manuscript is ready for publication in its current form and recommend to accept it.

Our response

We would like to thank Reviewer 3 for the comments and help in improving the manuscript.

Reviewer #4 (Remarks to the Author):

I read Sotiropoulos et al.'s manuscript with great interest. Here are my suggestions for improving it:

Reviewer's Comment 4.1

1/ Title: Do the study really show that wheat mildew co-migrated with humans?

The abstract reads: "After it spread throughout Eurasia, colonization brought it to America, where it hybridized with unknown grass mildew species. Recent trade brought USA strains to Japan, and European strains to China."

I agree that is likely human migration that introduced the pathogen in the Americas, but regarding China or Japan, trade is not migration.

I understand that the authors wanted a catchy title, but it shouldn't be misleading.

Our response

The reviewer is correct in that the title makes a too general statement. On one hand, we indeed show that the spread of wheat mildew aligns with historical human migration routes to Europe and America. On the other hand, we describe recent trade as a vehicle for pathogen spread. To include both aspects, we modified the title, using more precise language, as follows: "Global genomic analyses of wheat powdery mildew reveal association of pathogen spread with historical human migration and trade".

Reviewer's Comment 4.2

2/ Research presentation: the reading of the results section remains hampered by the use of redundant statistics, by the use of too many "see below", or by hypotheses that are not well formulated.

There are too many "see below" in the results section (n=10 occurrences). Consequently, some sentences do not describe results from the paragraph in which they are placed, but results from a subsequent paragraph, which really hinders reading.

Our response

We have removed two sentences that refer to later results which are not essential for the understanding of the text at this point (Lines 175-177 and 181-182). This removed redundant statements to statistics and analyses.

The three references to "below" text in the methods were considered non-essential and also removed. We kept two occurrences of "see below" where we consider it necessary for the understanding of the narrative, but inserting extensive additional results would divert the flow of the text (i.e. where the

reader might otherwise wonder if respective analyses were actually performed, Line 204 and Line 250).

One additional “see below” had to be inserted, since we previously referred to the section titled “conclusions”, which the journal format does not allow (Line 371).

We revised the formulation of three of the hypotheses that introduce paragraphs (see also responses to comments 4.5 and 4.8, Lines 207-209, Line 254, Line 332).

Reviewer’s Comment 4.3

L178: I don’t get why the authors wrote that “Chinese isolates (...) formed multiple subgroups with some showing similarity to European isolates”. First, I don’t see multiple subgroups, I see a gradient and two dots outside the main cloud. Second, China and Europe are clustering together along PCo2, not PCo1; is it sufficient to conclude that Chinese isolates “show some similarity to European isolates”?

Our response

This statement was removed as it also mostly refers to results of the ADMIXTURE analysis that are described in more detail later in the text (thereby also removing one “see below” reference, in response to comment 4.2).

Reviewer’s Comment 4.4

L193-200: Admixture results should be described without making reference to directionality. For instance, there’s no reason to write that “two Chinese isolates showed a high proportion of Israeli ancestry », as it suggests that Chinese isolates are – at least in part- from Israeli origin, while it is not what the analysis shows. The analysis shows that two Chinese isolates share ancestry with Israeli isolates, but it doesn’t tell whether migration was from China to Israel or the reverse. Same comment about Japanese isolates.

Our response

We now use more neutral language, stating simply that two Chinese isolates share a high proportion of ancestry with isolates from Israel, and that Japanese isolates share ancestry with mildews from East Asia and the USA (Lines 200-203).

However, in the case of the Japanese mildews, we provide historical information later in the text that wheat lines were actually introduced from the USA to Japan, demonstrating directionality of this ancestry.

Reviewer’s Comment 4.5

L202-206: Wind is not the only factor that can generate an association between genetic distance and geographic distance, so the topic sentence of this paragraph sounds awkward. Humans, or their *Helicobacter pylori* pathogens, were not dispersed by wind, but still, there is a correlation between genetic distance and geographic distance in these organisms. doi:10.1038/nature05562

Our response

We agree that factors other than wind dispersal can associate genetic and geographic distance. We therefore re-phrased the topical sentence to us testing the hypothesis that, if mildew is spread efficiently by wind across and between continents, we would find only low levels of association between genetic and geographical distances (Lines 207-209). Thus, our finding of strong association between genetic and geographical distances indicates that wind dispersal is not a major factor in the spread of the pathogen. We added a short statement in Lines 213-214.

Reviewer’s Comment 4.6

L222-228: Nucleotide diversity (π) is defined as the average number of nucleotide differences per site between pairs of DNA sequences within a population. What is the difference between π and the so called “average differences within population (ADW)” that the authors computed? If these two statistics are effectively different, what does “ADW” capture that π doesn’t? Justification needed.

Our response

Here, we used simple mismatch, Bray-Curtis, and Nei distance for measuring dissimilarity between the individual profiles, and therefore, the ADW metric with regard to each of them was applied. ADW diversity with regard to the simple mismatch dissimilarity between DNA sequences and the nucleotide

diversity (π) are, in principle, equivalent estimates of diversity within a population. However, in the case shown here, π was calculated as an average of all 10 kb genomic windows, while ADW was calculated using two different SNP sets (LD-pruned, and non-LD pruned, see also comment 4.7). For LD-pruned SNPs, ADW does capture lower simple mismatch distance (SMD) in *B. g. dicocci*, which π does not capture (see comment 4.7).

Additionally, ADW was needed to subsequently calculate distance of average differences (DAD). The use of DAD was recommended by reviewer 3 as an appropriate measure to capture the ‘net’ differences between populations (see our response to comment 4.10). We, therefore, found it appropriate to also display ADW values in figure 2b.

ADW diversity metric can be calculated with regard to different measures of dissimilarity between individual (DNA) profiles.

Reviewer’s Comment 4.7

L226: one has to read the methods section to understand what “filtered” and “unfiltered” mean. And why do these statistics need to be computed with two different datasets? Justification needed.

Our response

The main text was not clear in this case: initial quality filtering (for mapping quality and genotyping rate >90%) was applied to all SNP data from the start. However, in the statement the reviewer points out, we intended to refer to the use of LD-pruned and non-LD-pruned SNPs. This is now stated explicitly, replacing the imprecise terms “filtered” and “unfiltered”.

We computed the ADW statistics with LD-pruned as well as non-LD-pruned SNPs to check for consistency of results, since ADW values can be affected strongly by population structure. We added a statement to the methods, explaining this rationale for the use of the two datasets in the methods section (Lines 633-634). Indeed, one such difference can be seen for *B. g. dicocci* (which we had addressed already in Supplementary Fig. 9).

Reviewer’s Comment 4.8

L234: what is the point of constructing networks from several types of SNP sets? Do the authors expect differences in the historical signal captured by these different sets? Justification needed.

Our response

This is also in response to the previous point, we used multiple different datasets to check for consistency of results. Too stringent filtering can lead to loss of signals, while too relaxed filtering can introduce noise. We wanted to demonstrate that results were consistent across the range of parameters used by us here.

We, therefore, considered it appropriate to show results from different SNP datasets. This was also done in response to a previous comment of reviewer 3 who suggested using multiple distance calculations. We explain this now explicitly in the methods section (Lines 618-619).

Reviewer’s Comment 4.9

L248: “to study the colonization history of wheat mildew” could be the start of any paragraph.

Our response

We agree that the introductory sentence is too imprecise. We now use more precise language stating that we study “diversification history of wheat mildew populations” (Line 254).

Reviewer’s Comment 4.10

L260: I’m aware that it was proposed by one of the reviewers, but I don’t see the point of computing the “distance of average differences” other than to increase number of citations. I’m not a newbie in fungal population genetics but it’s the first time I come across this statistic. I know that d_{xy} and F_{st} are complementary (doi: 10.1111/mec.12796); but what does DAD capture that d_{xy} or F_{st} don’t? Justification needed.

Our response

F_{ST} measures difference between the total variability within a pooled population and the average variability within each population relative to the total variability. It can be used just as an approximate extent of differentiation of populations because the index does not reach its maximum value 1 for any two absolutely different unfixed populations with no shared alleles. Moreover, it is frequently utilized

for measuring pairwise distances between populations, although it is a relative estimate and does not actually measure the absolute difference between two populations. In our communication with reviewer 3, he argued that the d_{XY} statistic is inappropriate for measuring differentiation or distance between two populations because $d_{XX} > 0$ and thus distinguishes between two identical populations $X = Y$ (unless they both are fixed to a same allele at each locus). In contrast, the distance of average differences (DAD), is intended to capture only the 'net' differences between populations and is, thus, a measure for the absolute difference between two populations. DAD measures the difference between the average number of pairwise differences per site between individual profiles from populations X and Y and the average value of variability (pairwise differences per site) within each population. Therefore, $DAD_{XX} = 0$ for two identical populations $X = Y$, unlike d_{XX} (again, unless they both are fixed to a same allele at each locus). This is illustrated in the network graphs in Supplementary Figure 14 where DAD indeed shows more pronounced differentiation between populations. We added a statement to the methods giving the rationale for the use of DAD in addition to F_{ST} (Line 643-644).

Reviewer's Comment 4.11

L266: "In contrast, the Eurasian *B.g. tritici* populations (EUR-RUSK-ISR-CHN) showed more gradual separations from each other (Fig. 3b)." This suggest that you are comparing Eurasian populations to something? What does "gradual separation" mean? How can you tell separation is gradual on this plot?

Our response

Indeed, we wanted to state that the Eurasian populations separated individually at different time points from each other, in comparison to their rapid separation from USA and ARG populations, and the near-complete separation from the *B.g. dicocci* population. We changed the statement accordingly. We also replaced "gradual separation" with the more precise description that they separated more slowly from each other over time (Lines 273-275).

Reviewer's Comment 4.12

L267 - 271: Alternatively, as you indicated elsewhere, it could just be that the differentiation between ARG and USA was inflated by a founder event?

Our response

We now also mention this possibility at this point in the text (Lines 279-280).

Reviewer's Comment 4.13

L281: there was no wheat in the Americas 2,000-5,000 mildew generations ago (if generation time= 1 year).

Our response

The reviewer is correct. We were not explicit enough in explaining that the drop in N_e 2,000-5,000 generations ago likely reflects a bottleneck in the grass mildews with which the European mildews hybridized in America. That means, the mildews from (yet unknown) grasses may have experienced an early bottleneck. This could, for example, be the case for mildews that grew on barley relatives that were cultivated by Native Americans thousands of years ago. We now state this more clearly in the text (Lines 291-295).

Reviewer's Comment 4.14

Fig 2: too many digits in stats related to Mantel tests.

For the first row of Panel a plots, x-axis labels could be removed. For the second and third columns, y-axis labels could be removed.

Our response

We have modified the figure to consistently show only 6 digits and have removed the labels, as suggested by the reviewer.

Reviewer's Comment 4.15

Fig. 2b: histograms should be avoided.

See <https://journals.plos.org/plosbiology/article?id=10.1371/journal.pbio.1002128>

Our response

The publication the reviewer refers to is on the display of continuous data, which (we fully agree) should be presented, for example, in box plots that show individual data points. However, in this case, the bars are simply representation of single values, one calculated for each population. This data could be presented as a table, but we found it easier to see differences between populations in form of a bar plot.

Reviewer's Comment 4.16

Figure 2b: what does "Average" stand for? The average across all window-specific values? The average of population specific averages?

Our response

The word "average" refers to average across all 10 kb genomic windows. It is now stated in the Fig. 2b legend.

Reviewer's Comment 4.17

Fig 2 caption: Capitalize Mantel https://en.wikipedia.org/wiki/Nathan_Mantel

Moreover, what does "results below it" mean? Is it the Mantel test statistic r ? I think the correct caption is Supp Fig 7's.

Our response

Mantel is now capitalized, and the correct description (as in Supplementary Fig. 7) is now given.

Reviewer's Comment 4.18

3/ Literature cited: the authors fail to cite and discuss important papers investigating the population genetics of introduced fungal pathogens.

For instance, McMullan et al. 10.1038/s41559-018-0548-9 is not cited.

Our response

Hymenoscyphus fraxineus is indeed a particularly good example for haplotypes that were likely spread by trade, founding a new European pathogen population. We were not aware of this publication and now added the reference and a brief discussion statement (Lines 426-428).

Reviewer's Comment 4.19

Also, the authors cite papers that describe the population genetics of the chestnut blight pathogen in Britain or Tyrol instead of citing papers that investigated the population structure at larger scale, such as Stauber et al. 10.7554/eLife.56279 or Dutech et al. 10.1111/j.1365-294X.2012.05575.x

Our response

We have added the two citations to the introduction (Line 127).

Reviewer's Comment 4.20

The authors cite LeBlanc et al. 2019 instead of LeBlanc et al. 2021 10.1094/PHYTO-06-20-0219-FI

Our response

We replaced the LeBlanc et al. 2019 citation with the more recent LeBlanc et al. 2021.

Reviewer's Comment 4.21

L123 : Magnaporthae -> Magnaporthe. By the way, it would be unfair not to cite <https://doi.org/10.1186/s12915-020-00818-z> and <https://doi.org/10.1128/mBio.01806-17> in addition to, or in place of, Zhong et al, since these papers used more representative sets of isolates, and more detailed analyses.

Our response

The name *Magnaporthe* was corrected and we also now added the two citations in addition to Zhong et al (Lines 126-127).

Reviewer's Comment 4.22

4/ Mutation/substitution rate:

Experimental evolution doesn't measure the substitution rate, but the mutation rate (L577). The substitution rate is estimated in calibrated phylogenies along longer time spans. Rates of molecular evolution measured over shorter time spans tend to be overestimated, as some sites will be removed over time by negative selection. The rate presented here is more similar to a spontaneous mutation rate, than an evolutionary substitution rate.

The authors should be aware that several studies estimated substitutions rates in fungi, using tip dating in particular (e.g. *Cryptococcus gattii*, *Magnaporthe oryzae*, *Schizophyllum commune*). Estimates of the substitution rates are generally smaller than what is presented here (10-8/site/year, vs 10-7/site/generation here).

Would their conclusions regarding wheat - wheat mildew coevolution still hold with a different substitution rate? (L255-259)

This is really worth discussing.

Our response

The reviewer is correct and we have replaced "nucleotide substitution rate" with "mutation rate" throughout the manuscript. For the initial version of this manuscript, the mutation rate was used to date branches in phylogenetic trees. However, the trees were replaced with phylogenetic networks in this revised version (and the mutation rate was not needed anymore).

For this revised version, the mutation rate was only used to calculate effective population sizes and cross-coalescence shown in Fig 3. There, the application of the mutation rate indeed has an influence on timing of events.

We are aware that there is a level of uncertainty, and that that the measured rate may be over-estimated by maximum a factor of 2 to 5 (which brings it into the range of published evolutionary rates, e.g. Kasuga et al. 2002. *Mol Biol Evol* 19: 2318–2324, Chandler et al. 2018 *mSphere* 3:e00499-17).

Assuming a rate lower by a factor of 2-5 would, on one hand, affect the more speculative conclusion that a very recent bottleneck could be caused by modern breeding. On the other hand, it supports the common notion of the early separation of *B.g. dicocci* ~10,000 years ago.

Thus, the general conclusions regarding wheat/mildew co-evolution are not affected. Furthermore, our findings fit in with previous studies where different methods of molecular dating were used in mildew (*Fungal Genet Biol.* 2011, 48:327-34, *Nat Genet.* 2013, 45:1092-6).

Nevertheless, we now use more cautious language on the recent bottleneck (Lines 282-283) and added a note to the methods section stating that the mutation rate measured in the lab is likely higher than the evolutionary substitution rate due to negative selection over time in nature (Lines 603-605).

5/ Other comments:

Reviewer's Comment 4.23

Fig1a: I can't spot the purple and grey dots, as they are too small, or colors are not appropriate. Do you really need to plot *T. turgidum* and *Triticum* spp?

Our response

The reviewer is correct that it is not easy to spot the purple and grey dots. Grey was replaced by black for better visibility and the purple dot of *Triticum spp.* was replaced with red, since it was most likely collected from *Triticum aestivum*. This is also specifically mentioned in the Supplementary Table 1 with a (p).

Reviewer's Comment 4.24

L185: what is between parentheses should be a sentence. I don't see why it is between parentheses.

Our response

The parentheses were removed and the sentence connected to the previous one.

Reviewer's Comment 4.25

L199 delete “. K”.

Our response

The “. K” was deleted.

Reviewer's Comment 4.26

Supp Figure 12: selected isolates should be highlighted.

Our response

The selected isolates are now highlighted in Supplementary Fig. 12.